# Spotting false news and doubting true news: a systematic review and meta-analysis of news judgements

**Jan Pfänder** [1] **& Sacha Altay** [2] ✉

How good are people at judging the veracity of news? We conducted a systematic literature review and pre-registered meta-analysis of 303 effect sizes from 67 experimental articles evaluating accuracy ratings of true and fact-checked false news ($N_{\text{Participants}}$ = 194,438 from 40 countries across 6 continents). We found that people rated true news as more accurate than false news (Cohen's $d$ = 1.12 [1.01, 1.22]) and were better at rating false news as false than at rating true news as true (Cohen's $d$ = 0.32 [0.24, 0.39]). In other words, participants were able to discern true from false news and erred on the side of skepticism rather than credulity. We found no evidence that the political concordance of the news had an effect on discernment, but participants were more skeptical of politically discordant news (Cohen's $d$ = 0.78 [0.62, 0.94]). These findings lend support to crowdsourced fact-checking initiatives and suggest that, to improve discernment, there is more room to increase the acceptance of true news than to reduce the acceptance of fact-checked false news.

Many have expressed concerns that we live in a 'post-truth' era and that people cannot tell the truth from falsehoods anymore. In parallel, populist leaders around the world have tried to erode trust in the news by delegitimizing journalists and the news media[1]. Since the 2016 US presidential election, our systematic literature review shows that over 4,000 scientific articles have been published on the topic of false news. Across the world, numerous experiments evaluating the effect of interventions against misinformation or susceptibility to misinformation have relied on a similar design feature: having participants rate the accuracy of true and fact-checked false headlines, typically in a Facebook-like format, with an image, title, lede and source, or as an isolated title/claim. Taken together, these studies allow us to shed some light on the most common fears voiced about false news, namely, that people may fall for false news, distrust true news, or may be unable to discern true from false news. In particular, we investigated whether people rate true news as more accurate than fact-checked false news (discernment) and whether they were better at rating false news as inaccurate than at rating true news as accurate

(skepticism bias). We also tested various moderators of discernment and skepticism bias such as political congruence, the topic of the news, or the presence of a source.

Establishing whether people can spot false news is important to design interventions against misinformation. If people lack the skills to detect false news, interventions should focus on improving these skills. However, if people have the ability to spot false news yet buy into it anyway, the problem lies elsewhere and may be one of motivation or (in)attention that educational interventions will struggle to address.

Past work has reliably shown that people do not fare better than chance at detecting lies because most verbal and non-verbal cues people use to detect lies are unreliable[2]. Why would this be any different for detecting false news? People make snap judgements to evaluate the quality of the news they come across[3] and rely on seemingly imperfect proxies such as the source of information, police and fonts, the presence of hyperlinks, the quality of visuals, ads, or the tone of the text[4,5]. In experimental settings, participants report relying on intuitions and tacit knowledge to judge the accuracy of

¹Institut Jean Nicod, Département d'études cognitives, ENS, EHESS, PSL University, CNRS, Paris, France. ²Department of Political Science, University of Zurich, Zürich, Switzerland. ✉e-mail: sacha.altay@gmail.com

news headlines[6]. Yet, a scoping review of the literature on belief in false news (including a total of 26 articles) has shown that, in experiments, participants 'can detect deceitful messages reasonably well'[7]. Similarly, a survey on 150 misinformation experts has shown that 53% of experts agreed that 'people can tell the truth from falsehoods', while only 25% of experts disagreed with the statement[6]. Unlike the unreliable proxies people rely on to detect lies in interpersonal contexts, there are reasons to believe that some of the cues people use to detect false news may, on average, be reliable. For instance, the news outlets people trust the least do publish lower-quality news and more false news, as people's trust ratings of news outlets correlate strongly with fact-checkers' ratings in the United States and Europe[8,9]. Moreover, false news has some distinctive properties, such as being more politically slanted[10], being more novel, surprising, or disgusting, being more sensationalist, funnier, less boring and less negative[11,12], or being more interesting-if-true[13]. These features aim at increasing engagement, but they do so at the expense of accuracy and in many cases, people may pick up on it. This led us to pre-register the hypothesis that people would rate true news as more accurate than false news. Yet, legitimate concerns have been raised about the lack of data outside of the United States, especially in some Global South countries where the misinformation problem is arguably worse. Our meta-analysis covers 40 countries across 6 continents and directly addresses concerns about the over-representation of US data.

## People rate true news as more accurate than false news

While many fear that people are exposed to too much misinformation, too easily fall for it and are overly influenced by it, a growing body of researchers is worried that people are exposed to too little reliable information, commonly reject it and are excessively resistant to it[14,15]. Establishing whether true news skepticism (excessively rejecting true news) is of similar magnitude as false news gullibility (excessively accepting false news) is important for future studies on misinformation: if people are excessively gullible, interventions should primarily aim at fostering skepticism, whereas if people are excessively skeptical, interventions should focus on increasing trust in reliable information. For these reasons, in addition to investigating discernment (H1), we also looked at skepticism bias (H2) by comparing the magnitude of true news skepticism to false news gullibility. Research in psychology has shown that people exhibit a 'truth bias'[16,17], a tendency to accept incoming statements rather than to reject them. Similarly, work on interpersonal communication has shown that, by default, people tend to accept communicated information[18]. However, there are reasons to think that the truth-default theory may not apply to news judgements. It has been hypothesized that people display a truth bias in interpersonal contexts because information in these contexts is, in fact, often true[16]. When it comes to news judgements, it is not clear whether people by default expect news stories to be true. Trust in the news and journalists is low worldwide[19], and a substantial part of the population holds cynical views of the news[20]. Similarly, populist leaders across the world have attacked the credibility of the news media and instrumentalized the concept of fake news to discredit quality journalism[21,22]. Disinformation strategies such as 'flooding the zone' with false information[23,24] have been shown to increase skepticism in news judgements[6]. Moreover, in many studies included in our meta-analysis, the news stories were presented in a social media format (most often Facebook), which could fuel skepticism in news judgements. People trust news[3]—and information more generally[25]—less on social media than on news websites. In line with these observations, some empirical evidence suggests that for news judgements, people display the opposite of a truth bias[26], namely, a skepticism bias, and view true news as false[6,27,28]. We thus predicted that when judging the accuracy of news, participants will err on the side of skepticism more than on the side of gullibility.

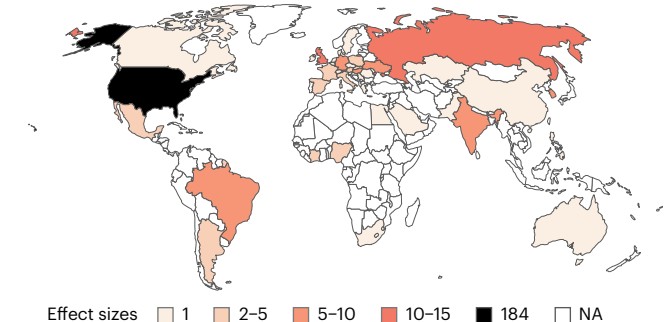

**Fig. 1** | Map of the number of effect sizes per country. Countries are coloured based on the number of effect sizes. The darker a country, the more effect sizes. Countries in white are not covered in the sample.

## People are better at rating false news as false than true news as true

Finally, we investigated potential moderators of H1 and H2, such as the country where the experiment was conducted, the format of the news headlines, the topic, whether the source of the news was displayed, and the political concordance of the news. Past work suggests that displaying the source of the news has a small effect at best on accuracy ratings[29], whereas little work has investigated differences in news judgements across countries, topics and formats. The effect of political concordance on news judgements is debated. Participants may be motivated to believe politically congruent (true and false) news, motivated to disbelieve politically incongruent news, or not be politically motivated at all but still display such biases[30]. We formulated research questions instead of hypotheses for our moderator analyses because of a lack of strong theoretical expectations.

## Results
### Descriptives
We conducted a systematic literature review and pre-registered the meta-analysis based on 67 publications, providing data on 195 samples (194,438 participants) and 303 effects (that is $k$, the meta-analytic observations). Our meta-analysis includes publications from 40 countries across 6 continents. However, 34% of all participants were recruited in the United States alone, and 54% in Europe. Only 6% of participants were recruited in Asia, and even less in Africa (2%; see Fig. 1 for the number of effect sizes per country). The average sample size was 997.12 (minimum = 19, maximum = 32,134, median = 482).

In total, participants rated the accuracy of 2,167 unique news items. On average, a participant rated 19.76 news items per study (min. = 2, max. = 240, median = 18). For 71 samples, news items were sampled from a pool of news (the pool size ranged from 12 to 255, with an average pool size of 57.46 items). The vast majority of studies (294 out of 303 effects) used a within-participant design for manipulating news veracity, with each participant rating both true and false news items. Almost all effect sizes are from online studies (286 out of 294).

### Analytic procedures
All analyses were pre-registered unless explicitly stated otherwise (for deviations, see Methods). The choice of models was informed by simulations we conducted before having the data. To test H1, we calculated a discernment score by subtracting the mean accuracy ratings of false news from the mean accuracy ratings of true news, such that higher scores indicate better discernment. This differential measure of discernment is common in the literature on misinformation[31]. To test H2, we first calculated a judgement error for true and false news. Error is defined as the distance between optimal accuracy ratings and actual accuracy ratings (Fig. 2). We then calculated the skepticism bias as the difference between the two errors, subtracting the false

news error score from the true news error score. Note that we cannot use more-established Signal Detection Theory (SDT) measures because we rely on mean ratings and not individual ratings. However, in Supplementary Section H, we show that for the studies we have raw data on, our main findings hold when relying on $d'$ (sensitivity) and $c$ (response bias) from SDT.

To be able to compare effect sizes across different scales, we calculated Cohen's $d$, a common standardized mean difference. To account for statistical dependence between true and false news ratings arising from the within-participant design used by most studies (294 out of 303 effect sizes), we calculated the standard error following the Cochrane recommendations for crossover trials[32]. For the remaining 9 effect sizes from studies that used a between-participant design, we calculated the standard error assuming independence between true and false news ratings (see Methods). In Supplementary Section A, we show that our results hold across alternative standardized effect measures, including the one we originally pre-registered—a standardized mean change using change score standardization (SMCC). We chose to deviate from the pre-registration and use Cohen's $d$ instead, because it is easier to interpret and corresponds to the standards recommended by the Cochrane manual[32]. In Supplementary Section A, we also provide effect estimates in units of the original scales separately for each scale.

We used multilevel meta-models with clustered standard errors at the sample level to account for cases in which the same sample contributed various effect sizes (that is, the meta-analytic units of observation). All confidence intervals reported in this paper are 95% confidence intervals. All statistical tests are two-tailed.

## Main results

**Discernment.** Supporting H1, participants rated true news as more accurate than false news on average. Pooled across all studies, the average discernment estimate is large ($d = 1.12$ [1.01, 1.22], $z = 20.79$, $P < 0.001$). As shown in Fig. 3, 298 of 303 estimates are positive. Of the positive estimates, three have a confidence interval that includes zero, as does one of the negative estimates. Most of the variance in the effect sizes observed above is explained by between-sample heterogeneity ($I2_{between} = 92.04\%$). Within-sample heterogeneity is comparatively small ($I2_{within} = 7.93\%$), indicating that when the same participants were observed on several occasions (that is, the same sample contributed several effect sizes), on average, discernment performance was similar across those observations. The share of the variance attributed to sampling error is very small (0.03%), which is indicative of the large sample sizes and thus precise estimates.

**Skepticism bias.** We found support for H2, with participants better able to rate false news as inaccurate than to rate true news as accurate (that is, false news discrimination was on average higher than true news discrimination). However, the average skepticism bias estimate is small ($d = 0.32$ [0.24, 0.39], $z = 8.11$, $P < 0.001$). As shown in Fig. 3, 203 of 303 estimates are positive. Of the positive estimates, six have a confidence interval that includes zero, as do seven of the negative estimates. By contrast with discernment, most of the variance in skepticism bias is explained by within-sample heterogeneity ($I2_{within} = 60.96\%$; $I2_{between} = 38.99\%$; sampling error = 0.05%). Whenever we observe within-sample variation in our data, it is because several effects were available for the same sample. This is mostly the case for studies with multiple survey waves, or when effects were split by different news topics, suggesting that these factors may account for some of that variation. In the moderator analyses below, most variables vary between samples, thereby glossing over much of that within-sample variation, except for political concordance.

## Moderators

Following the pre-registered analysis plan, we ran a separate meta-regression for each moderator by adding the respective moderator

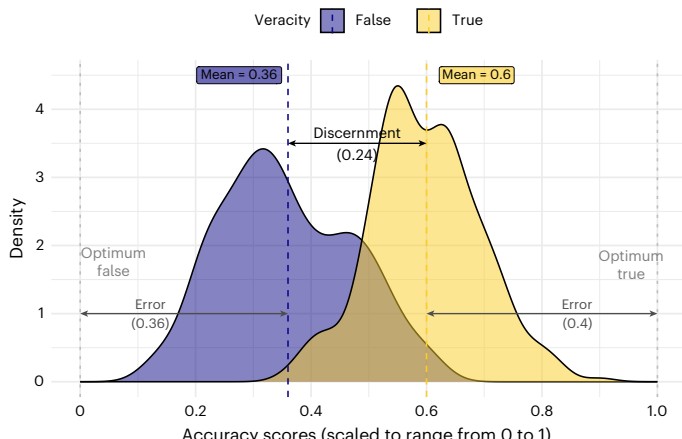

**Fig. 2 | Illustration of outcome measures.** Distributions of accuracy ratings for true and fact-checked false news, scaled to range from 0 to 1. The figure illustrates discernment (the distance between the mean for true news and the mean for false news) and the errors (distance to the right end for true news and to the left end for false news) from which the skepticism bias is computed. A larger error for true news compared with false news yields a positive skepticism bias. In this descriptive figure, unlike in the meta-analysis, ratings and outcomes sizes are not weighted by sample size.

variable as a fixed effect to the multilevel meta-models. We report regression tables and visualizations in Supplementary Section B. Here we report the regression coefficients as 'Deltas', since they designate differences between categories. For example, in the moderator analysis of political concordance on skepticism bias, 'concordant' marks the baseline category. The predicted value for this category can be read from the intercept (−0.2). The 'Delta' is the predicted difference between concordance and discordance (0.78). To obtain the predicted value for discordant news, one needs to add the 'Delta' to the intercept (−0.2 + 0.78 = 0.58).

**Cross-cultural variability.** For samples based in the United States (184/303 effect sizes), discernment was higher on average than for samples based in other countries (ΔDiscernment = 0.23 [0.02, 0.44], $z = 2.14$, $P = 0.033$; baseline discernment other countries pooled = 0.99 [0.84, 1.14], $z = 12.82$, $P < 0.001$). However, we did not find a statistically significant difference regarding skepticism bias (ΔSkepticism bias = 0.04 [−0.12, 0.19], $z = 0.47$, $P = 0.638$). A visualization of discernment (F1) and skepticism bias (F2) across countries can be found in Supplementary Section F.

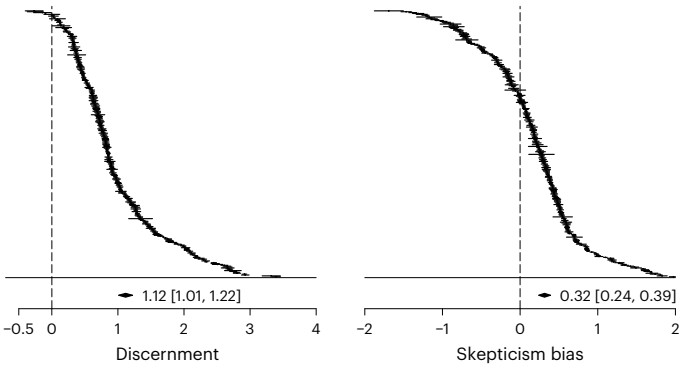

**Fig. 3 | Forest plots for discernment and skepticism bias.** All $n = 303$ effect sizes for both outcomes. Effects are weighted by their sample size. Effect sizes are calculated as Cohen's $d$. Horizontal bars represent 95% confidence intervals. The average estimate is the result of a multilevel meta model with clustered standard errors at the sample level.

**Scales.** The studies in our meta-analysis used a variety of accuracy scales, including both binary (for example, "Do you think the above headline is accurate? - Yes, No") and continuous ones (for example, "To the best of your knowledge, how accurate is the claim in the above headline? 1 = Not at all accurate, 4 = Very accurate").

Regarding discernment, two scale types differed from the most common 4-point scale (baseline discernment 4-point-scale = 1.28 [1.07, 1.49], $z = 11.96$, $P < 0.001$): both 6-point scales ($\Delta$Discernment = −0.41 [−0.7, −0.12], $z = −2.80$, $P = 0.006$) and binary scales ($\Delta$Discernment = −0.37 [−0.66, −0.08], $z = −2.50$, $P = 0.013$) yielded lower discernment. Regarding skepticism bias, studies using a 4-point scale (baseline skepticism bias 4-point scale = 0.51 [0.3, 0.72], $z = 0.75$, $P < 0.001$) reported a larger skepticism bias compared with studies using a binary or a 7-point scale ($\Delta$Skepticism bias = −0.29 [−0.51, −0.06], $z = −2.47$, $P = 0.014$ for binary scales; −0.50 [−0.76, −0.23], $z = −3.67$, $P < 0.001$ for 7-point scales). Interpreting these observed differences is not straightforward; we further discuss differences between binary and Likert-scale studies in Supplementary Section D.

**Format.** Studies using headlines with pictures as stimuli ($\Delta$Skepticism bias = 0.22 [0.04, 0.39], $z = 2.45$, $P = 0.015$; 65 effects), or headlines with pictures and a lede ($\Delta$Skepticism bias = 0.33 [0.14, 0.52], $z = 3.40$, $P < 0.001$; 56 effects), displayed a stronger skepticism bias compared with studies relying on headlines with no picture/lede (baseline skepticism bias headlines only = 0.23 [0.13, 0.33], $z = 4.45$, $P < 0.001$; 163 effects). We do not find differences related to format for discernment, neither for headlines with pictures ($\Delta$Discernment = −0.01 [−0.28, 0.27], $z = −0.04$, $P = 0.969$), nor for headlines with pictures and a lede ($\Delta$Discernment = 0.11 [−0.12, 0.33], $z = 0.93$, $P = 0.353$).

**Topic.** We did not find statistically significant differences in discernment and skepticism bias across news topics, when distinguishing between the categories 'political' ($\Delta$Skepticism bias = 0.03 [−0.13, 0.19], $z = 0.43$, $P = 0.671$; $\Delta$Discernment = −0.26 [−0.51, 0], $z = −1.98$, $P = 0.049$; 196 effects; 43 articles), 'covid-19' (baseline; 54 effects; 13 articles) and 'other' ($\Delta$Skepticism bias = −0.02 [−0.2, 0.16], $z = −0.22$, $P = 0.825$; $\Delta$Discernment = −0.01 [−0.35, 0.34], $z = −0.03$, $P = 0.976$; 53 effects; 20 articles), a category regrouping all news not explicitly labelled as 'covid-19' or 'political', including news about health, cancer, science, economics, history or military matters.

**Sources.** In line with past findings, we did not observe a statistically significant difference in discernment between studies that displayed the source of the news items ($\Delta$Discernment = −0.22 [−0.47, 0.03], $z = −1.75$, $P = 0.082$; 112 effects) and studies that did not (147 effects; for 44 effects, this information was not explicitly provided). We did not find a difference regarding skepticism bias either ($\Delta$Skepticism bias = 0.11 [−0.06, 0.29], $z = 1.30$, $P = 0.194$).

**Political concordance.** The moderators investigated above were (mostly) not experimentally manipulated within studies, but instead varied between studies, which impedes causal inference. Political concordance is an exception in this regard. It was manipulated within 31 different samples, across 14 different papers. In those experiments, typically, a pre-test establishes the political slant of news headlines (for example, pro-Republican vs pro-Democrat). Participants then rate the accuracy for news items of both political slants and provide information about their own political stance. The ratings of items are then grouped into concordant or discordant (for example, pro-Republican news rated by Republicans will be coded as concordant while pro-Republican news rated by Democrats will be coded as discordant).

Political concordance had no statistically significant effect on discernment ($\Delta$Discernment = 0.08 [−0.01, 0.17], $z = 1.72$, $P = 0.097$). It did, however, make a difference regarding skepticism bias (Fig. 4): when rating concordant items, there was no evidence that participants showed

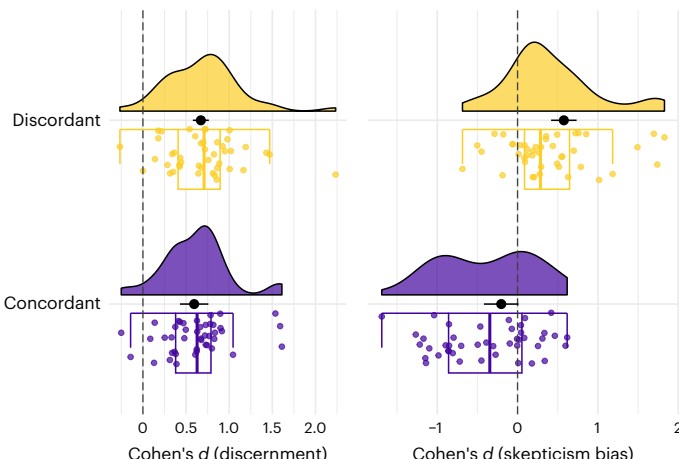

**Fig. 4 | Effect of political concordance on discernment and skepticism bias.** Distribution of the $n = 44$ effect sizes for politically concordant and discordant items. The black dots represent the predicted average of the meta-regression, the black horizontal bars the 95% confidence intervals. Note that the figure does not represent the different weights (that is, the varying sample sizes) of the data points, but these weights are taken into account in the meta-regression. For the boxplots, the box represents the interquartile range (IQR), that is, the distance between the first and third quartiles, the centre line indicates the median, and the outer lines (whiskers) extend to 1.5 times the IQR or the most extreme values within this range. Data beyond the end of the whiskers are considered 'outlying' points and are plotted individually.

a skepticism bias (baseline skepticism bias concordant items = −0.20 [−0.42, 0.01], $z = −1.93$, $P = 0.064$), while for discordant news items, participants displayed a positive skepticism bias ($\Delta$Skepticism bias = 0.78 [0.62, 0.94], $z = 10.04$, $P < 0.001$). In other words, participants were not gullible when facing concordant news headlines (as would have been suggested by a negative skepticism bias), but were skeptical when facing discordant ones.

## Individual-level data
In the results above, accuracy ratings were averaged across participants. It is unclear how these average results generalize to individuals. Do they hold for most participants? Or are they driven by a relatively small group of participants with either excellent discernment skills or extreme skepticism? For 22 articles ($N_{\text{Participants}} = 42{,}074$, $N_{\text{Observations}} = 813{,}517$), we have the raw data for all ratings that individual participants made on each news headline they saw. On these data, we ran a descriptive, non-pre-registered analysis: we calculated a discernment and skepticism bias score for each participant on the basis of all the news items they rated. To compare across different scales, we transposed all accuracy scores on a scale from 0 to 1, resulting in a range of possible values from −1 to 1 for both discernment and skepticism bias.

As shown in Fig. 5, 79.92% of individual participants had a positive discernment score, and 59.06% of participants had a positive skepticism bias score. Therefore, our main results based on mean ratings across participants seem to be representative of individual participants (see Supplementary Section C for further discussion).

## Discussion
This meta-analysis sheds light on some of the most common fears voiced about false news. In particular, we investigated whether people are able to discern true from false news and whether they are better at judging the veracity of true news or false news (skepticism bias). Across 303 effect sizes ($N_{\text{Participants}} = 194{,}438$) from 40 countries across 6 continents, we found that people rated true news as much more accurate than fact-checked false news ($d_{\text{discernment}} = 1.12$ [1.01, 1.22]) and are

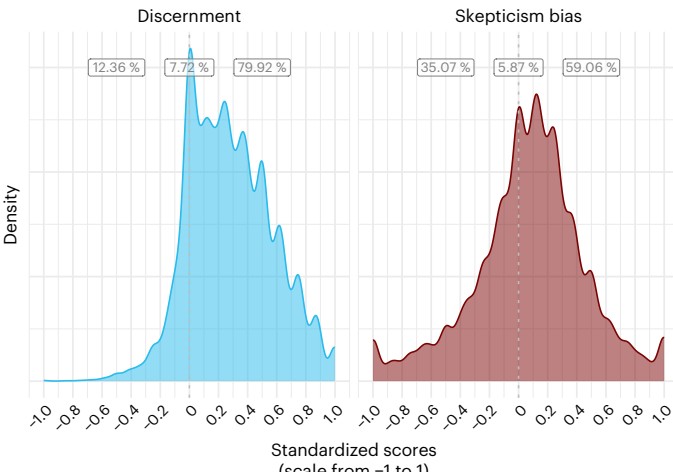

**Fig. 5 | Outcomes at the participant level.** Distribution of average discernment and skepticism bias scores of individual participants in the subset of studies that we have raw data on. We standardized original accuracy ratings to range from 0 to 1. The lowest possible score is −1 where, for discernment, an individual classified all news wrongly, and for skepticism bias, an individual classified all true news correctly (as true) and all false news incorrectly (as true). The highest possible score is 1 where, for discernment, an individual classified all news correctly, and for skepticism bias, an individual classified all true news incorrectly (as false) and all false news correctly (as false). The percentage labels (from left to right) represent the share of participants with a negative score, a score of exactly 0, and a positive score, for both measures.

slightly better at rating fact-checked false news as inaccurate than at rating true news as accurate ($d_{\text{skepticism bias}}$ = 0.32 [0.24, 0.39]).

The finding that people can discern true from false news when prompted to do so has important implications for interventions against misinformation. First, it suggests that most people do not lack the skills to spot false news (at least the kind of fact-checked false news used in the studies included in our meta-analysis). If people do not lack the skills to spot false news, why do they sometimes fall for false news? In some contexts, people may lack the motivation to use their discernment skills or may apply them selectively[33,34]. Thus, instead of teaching people how to spot false news, it may be more fruitful to target their motivations and incentives, either by manipulating features of the environment in which people encounter news[35,36], or by prompting people to use their critical thinking skills and pay more attention to accuracy[33]. For instance, it has been shown that design features of current social media environments sometimes impede discernment and not incentivize the sharing of accurate information[37].

Second, the fact that people can, on average, discern true from false news lends support to crowdsourced fact-checking initiatives. While fact-checkers cannot keep up with the pace of false news production, the crowd can, and it has been shown that even small groups of participants perform as well as professional fact-checkers[38,39]. The cross-cultural scope of our findings suggests that these initiatives may be fruitful in many countries worldwide. In every country included in the meta-analysis, participants on average rated true news as more accurate than false news (see Supplementary Section F). In line with past work[38], we have shown that this was not only true on average, but also for a large majority (79.92%) of participants for which we had individual-level data. Our results are also informative for the work of fact-checkers. Since people appear to be quite good at discerning true from false news, fact-checkers may want to focus on headlines that are less clearly false or true. However, we cannot rule out that people's current discernment skills stem in part from the current and past work of fact-checking organizations. Fact-checking remains important and complementary to, not in tension with, crowdsourcing efforts.

The fact that people disbelieve true news slightly more than they believe fact-checked false news speaks to the nature of the misinformation problem and how to fight it: the problem may be less that people are gullible and fall for falsehoods too easily, but instead that people are excessively skeptical and do not believe reliable information enough[15,40]. Even assuming that the rejection of true news and the acceptance of false news are of similar magnitude (and that both can be improved), given that true news are much more prevalent in people's news diet than false news[41], true news skepticism may be more detrimental to the accuracy of people's beliefs than false news acceptance[14]. This skepticism is concerning in the context of the low and declining trust and interest in news across the world[42], as well as the attacks of populist leaders on the news media[22] and growing news avoidance[43]. Interventions aimed at reducing misperceptions should therefore consider increasing the acceptance of true news in addition to reducing the acceptance of false news[14,44]. At the very least, when testing interventions, researchers should evaluate their effect on both true and false news, not just false news[45]. At best, interventions should use methods that allow estimation of discrimination while accounting for response bias, such as SDT, and make sure that apparent increases in discernment are not due to a more conservative response bias[28,46]. This is important given that recent evidence suggests that some interventions against misinformation, such as media literacy tips[47], fact-checking[48], or educational games aimed at inoculating people against misinformation[28], may reduce belief in false news at the expense of fostering skepticism towards true news.

We also investigated various moderators of discernment and skepticism bias. We found that discernment was greater in studies conducted in the United States compared with the rest of the world. This could be due to the inclusion of many countries from the Global South, where belief in misinformation and conspiracy theories has been documented to be higher[49]. In line with past work[29], the presence of a source had no statistically significant effects on discernment or skepticism bias. Neither did the topic of the news. Participants showed greater skepticism towards headlines presented in a social media format (with an image and lede) or along with an image, compared with plain headlines with just text. This suggests that the skepticism towards true news documented in this meta-analysis may be partially due to the social media format of the news headlines. Past work has shown that people report trusting news on social media less[3,19], and experimental manipulations have shown that the Facebook news format reduces belief in news[50,51], although the causal effects documented in these experiments are much smaller than the ones observed in surveys[52]. Low trust in news on social media may be a good thing, given that on average, news on social media may be less accurate than news on news websites, but it is also worrying since most of news consumption worldwide is shifting online and on social media in particular[43].

The political concordance of the news had no statistically significant effect on discernment, but participants were excessively skeptical of politically discordant news. That is, participants were equally skilled at discerning true from false news for concordant and discordant items, but rated all news as more false when it was politically discordant. This finding is in line with recent evidence on partisan biases in news judgements[53] and supports the idea that people are not excessively gullible of news they agree with, but are instead excessively skeptical of news they disagree with[15,54]. It suggests that interventions aimed at reducing partisan motivated reasoning, or at improving political reasoning in general, should focus more on increasing openness to opposing viewpoints than on increasing skepticism towards concordant viewpoints. Future studies should investigate whether the effect of congruence is specific to politics or if it holds across other topics, and compare it to a baseline of neutral items.

Our meta-analysis has two main conceptual limitations. First, participants evaluated the news stories in artificial settings that do not mimic the real world. For instance, the mere fact of asking participants

to rate the accuracy of the news stories may have increased discernment by increasing attention to accuracy[33]. When browsing on social media, people may be less discerning (and perhaps less skeptical) than in experimental settings because they would pay less attention to accuracy[37]. However, given people's low exposure to misinformation online[55], they may protect themselves from misinformation not by detecting it on the spot, but by relying on the reputation of the sources and avoiding unreliable sources[56]. Second, our results reflect choices made by researchers about news selection. The vast majority of studies in our meta-analysis relied on fact-checked false news, determined by fact-checking websites, such as Snopes or PolitiFact. By contrast, three papers[38,57,58] automated their news selection by scraping headlines from media outlets in real time. In these studies, both participants and fact-checkers (or the researchers themselves, in the case of ref. 57) rated the veracity of the headlines shortly after they got published. The three studies (53 effect sizes; 10,170 participants; all in the United States) find (1) lower discernment than our meta-analytic average and (2) a negative skepticism (that is, a credulity) bias (see Supplementary Section G for a detailed discussion). This highlights the importance of news selection in misinformation research: researchers need to think carefully about what population of news they sample from and be clear about the generalizability of their findings[40,59].

Furthermore, our meta-analysis has methodological limitations, which we address in a series of robustness checks in the Supplementary Sections. We show that our results hold across alternative effect size estimators (Supplementary Section A), and that the results are similar when running a participant-level analysis on a subset of studies for which we have raw data (Supplementary Section C) or when relying on $d'$ (sensitivity) and $c$ (response bias) from SDT for that subset (Supplementary Section H). A comparison of binary and Likert-scale ratings suggests that the skepticism bias stems partly from mis-classifications and partly from degrees of confidence (Supplementary Section D).

In conclusion, we found that in experimental settings, people are able to discern mainstream true news from fact-checked false news, but when they err, they tend to do so on the side of skepticism more than on the side of gullibility (although the effect is small and probably contingent on false news selection). These findings lend support to crowdsourced fact-checking initiatives and suggest that, to improve discernment, there may be more room to increase the acceptance of true news than to reduce the acceptance of false news.

## Methods

### Data

We undertook a systematic review and meta-analysis of the experimental literature on accuracy judgements of news, following the PRISMA guidelines[60] (Extended Data Fig. 1). All records resulting from our literature searches can be found on the OSF project page (https://osf.io/96zbp/). We documented rejection decisions for all retrieved papers (see OSF project page).

**Eligibility criteria.** For a publication to be included in our meta-analysis, we set six eligibility criteria: (1) We considered as relevant all document types with original data (not only published ones, but also reports, preprints and working papers). When different publications were using the same data, a common scenario, we included only one publication (which we picked arbitrarily). (2) We only included articles that measured perceived accuracy (including 'accuracy', 'credibility', 'trustworthiness', 'reliability' or 'manipulativeness') and (3) did so for both true and false news. (4) We only included studies relying on real-world news items. Accordingly, we excluded studies in which researchers made up the false news items, or manipulated the properties of the true news items. (5) We could only include articles that provided us with the relevant summary statistics (means and standard deviations for both false and true news), or publicly available data that allowed us to calculate those. In cases where we were not able to retrieve the relevant

summary statistics either way, we contacted the authors. (6) Finally, to ensure comparability, we only included studies that provided a neutral control condition. For example, ref. 61, among other things, tested the effect of an interest prime vs an accuracy prime. A neutral control condition—one that is comparable to those of other studies—would have had no prime at all. We therefore excluded this paper. Rejection decisions for all retrieved papers are documented and can be accessed on the OSF project page (https://osf.io/96zbp/). We provide a list of all included articles in Supplementary Section J.

**Deviations from eligibility criteria.** We followed our eligibility criteria with 4 exceptions. We rejected one paper on the basis of a criterion that we had not previously set: scale asymmetry. Reference 62 asked participants: "According to your knowledge, how do you rate the following headline?", providing a very asymmetrical set of answer options ('1–not credible; 2–somehow credible; 3–quite credible; 4–credible; 5–very credible'). The paper provides 6 effect sizes, all of which strongly favour our second hypothesis (one effect being as large as $d = 2.54$). We decided to exclude this paper from our analysis because of its very asymmetric scale (that is, there is no clear scale midpoint and the labels do not symmetrically map onto a false/true dichotomy, which contrasts with the other scales included here). Further, we stretched our criterion for real-world news on three instances: refs. 63,64 used artificial intelligence trained on real-world news to generate false news, and ref. 65 had journalists create the false news items. We reasoned that asking journalists to write news should be similar enough to real-world news, and that large language models (LLMs) already produce news headlines that are indistinguishable from real news, so it should not make a big difference.

**Literature search.** Our literature review is based on two systematic searches. We conducted our first search on 2 March 2023 using Scopus (search string: '"false news" OR "fake news" OR "false stor*" AND "accuracy" OR "discernment" OR "credibilit*" OR "belief" OR "susceptib*"') and Google Scholar (search string: '"Fake news"|"False news"|"False stor*"|"Accuracy"|"Discernment"|"Credibility"|"Belief"|"Suceptib*", no citations, no patents'). On Scopus, given the initially high volume of papers (12,425), we excluded papers not written in English, that were not articles or conference papers, and that were from disciplines that are probably irrelevant for the present search (for example, Dentistry, Veterinary, Chemical Engineering, Chemistry, Nursing, Pharmacology, Microbiology, Materials Science, Medicine) or unlikely to use an experimental design (for example, Computer Science, Engineering, Mathematics; see Supplementary Section I for detailed search string). After these filters were applied, we ended up with 4,002 results. The Google Scholar search was intended to identify important preprints or working papers that the Scopus search would have missed. We only considered the first 980 results of that search—a limit imposed by the 'Publish or Perish' software we used to store Google Scholar search results in a data frame.

After submitting a manuscript version, reviewers remarked that not including the terms 'misinformation' or 'disinformation' in our search string might have omitted relevant results. On 22 March 2024, we therefore conducted a second, pre-registered (https://doi.org/10.17605/OSF.IO/YN6R2, registered on 12 March 2024) search using an extended query string (search string for both Scopus and Google Scholar: ' "false news" OR "fake news" OR "false stor*" OR "misinformation" OR "disinformation") AND ("accuracy" OR "discernment" OR "credibilit*" OR "belief" OR "susceptib*" OR "reliab*" OR "vulnerabi*" '; see Supplementary Section I for detailed search string). After removing duplicates (642 between the first and the second Scopus search and 269 between the first and the second Google Scholar search), the second search yielded an additional 1,157 results for Scopus and 711 results for Google Scholar. In total, the Scopus searches yielded 5,159, and the Google Scholar searches 1,691 unique results.

We identified and removed 338 duplicates between the Google Scholar and the Scopus searches, and ended up with 6,512 documents for screening. We had two screening phases: first, titles, then abstracts. For the results from the second literature search, both authors screened the results independently. In case of conflicting decisions, an article passed onto the next stage (that is, received abstract screening or full text assessment). For the results from the second literature search, screening was done on the basis of titles and abstracts only, so that the screeners would not be influenced by information on the authors or the publishing journal. The vast majority of documents (6,248) had irrelevant titles and were removed during that phase. Most irrelevant titles were not about false news or misinformation (for example, "Formation of a tourist destination image: Co-occurrence analysis of destination promotion videos"), and some were about false news or misinformation but were not about belief or accuracy (for example, "Freedom of Expression and Misinformation Laws During the COVID-19 Pandemic and the European Court of Human Rights"). We stored the remaining 264 records in the reference management system Zotero for retrieval. Of those, we rejected a total of 217 papers that did not meet our inclusion criteria. We rejected 87 papers on the basis of their abstract and 130 after assessment of the full text. We documented all rejection decisions (available on the OSF project page, https://osf.io/96zbp/). We included the remaining 47 papers from the systematic literature search. To complement the systematic search results, we conducted forward and backward citation search through Google Scholar. We also reviewed additional studies that we had on our computers and papers we found via Twitter (mostly working papers). Taken together, we identified an additional 47 papers via those methods. Of these, we excluded 27 papers after full text assessment because they did not meet our inclusion criteria. For these papers, we also documented our exclusion decisions, which can be found together with the ones for the systematic search on the OSF project page (https://osf.io/96zbp/). We included the remaining 20 papers. In total, we included 67 papers in our meta-analysis[6,8,12,13,26,29,34,37,44,53,57–59,63–116]: 47 peer reviewed and 20 grey literature (reports and working papers). We retrieved the relevant summary statistics directly from the paper for 21 papers, calculated them ourselves on the basis of publicly available raw data for 31 papers, and got them from the authors after request for 15 papers.

### Statistical methods

Unless stated otherwise, all the analyses were pre-registered 28 April 2023 (https://doi.org/10.17605/OSF.IO/SVC7U). Our choice of statistical models was informed by simulations, which can also be found on the OSF project page. We conducted all analyses in R v.4.2.2 (31 October 2022)[117] using Rstudio (v.2024.9.0.375)[118] and the tidyverse package (v.2.0.0)[119]. We relied on the functions escalc() for effect size calculations, rma.mv() for models, and robust() for clustered standard errors, all from the metafor package (v.4.6.0)[120].

**Deviations from pre-registration.** We pre-registered standardized mean changes using change score standardization (SMCC) as an estimator for our effect sizes[121]. However, in line with the Cochrane guidelines[32], we chose to rely on the more common Cohen's *d* for the main analysis. We report results from the pre-registered SMCC (along with other alternative estimators) in Supplementary Section A. All estimators yielded similar results. We did not pre-register considering scale symmetry, and proportion of true news and false news selection (taken from fact-checking sites vs verified by researchers) as moderator variables. We report the results regarding these variables in Supplementary Section B.

**Outcomes.** We have two complementary measures of assessing the quality of people's news judgement. The first measure is discernment. It measures the overall quality of news judgement across true and false news. We calculated discernment by subtracting the mean accuracy ratings of false news from the mean accuracy ratings of true news, such that more positive scores indicate better discernment. However, discernment is a limited diagnostic of the quality of people's news judgement. Imagine a study A in which participants rate 50% of true news and 20% of false news as accurate and a study B rating 80% of true news and 50% of false news as accurate. In both cases, the discernment is the same: participants rated true news as more accurate by 30 percentage points than false news. However, the performance by news type is very different. In study A, people do well for false news—they only mistakenly classify 20% as accurate—but are at chance for true news. In study B, it is the opposite. We therefore used a second measure: skepticism bias. For any given level of discernment, it indicates whether people's judgements were better for true news or for false news and to what extent. First, we calculated an error for false and true news separately, which we defined as the distance of participants' actual ratings to the best possible ratings. For example, for study A, the mean error for true news is 50% (100%−50%), because in the best possible scenario, participants would have classified 100% of true news as true. The error for false news in Study A is 20% (20%−0%), because the best possible performance for participants would have been to classify 0% of false news as accurate. We calculated skepticism bias by subtracting the mean error for false news from the mean error for true news. For example, for Study A, the skepticism bias is 30% (50%−20%). A positive skepticism bias indicates that people doubt true news more than they believe false news.

Skepticism bias can only be (meaningfully) interpreted on scales using symmetrical labels, that is, the intensity of the labels to qualify true and false news are equivalent (for example, 'True' vs 'False' or 'Definitely fake' [1] to 'Definitely real' [7]). Of the effects included in the meta-analysis, 69% used scales with perfectly symmetrical labels, while 26% used imperfectly symmetrical scale labels, that is, the intensity of the labels to qualify true and false news are similar but not equivalent (for example, [1] not at all accurate, [2] not very accurate, [3] somewhat accurate, [4] very accurate; here for instance 'not at all accurate' is stronger than 'very accurate'). We could only compute symmetry for scales that explicitly labelled scale points, resulting in missing values for 5% of effects. In Supplementary Section B, we show that scale symmetry has no statistically significant effect on skepticism bias.

**Effect sizes.** The studies in our meta-analysis used a variety of response scales, including both binary (for example, "Do you think the above headline is accurate? - Yes, No") and continuous ones (for example "To the best of your knowledge, how accurate is the claim in the above headline? 1 = Not at all accurate, 4 = Very accurate"). To be able to compare across the different scales, we calculated standardized effects, that is, effects expressed in units of standard deviations. Precisely, we calculated Cohen's *d* as

$$\text{Cohen's } d = \frac{\bar{x}_{\text{true}} - \bar{x}_{\text{false}}}{\text{SD}_{\text{pooled}}} \quad (1)$$

with

$$\text{SD}_{\text{pooled}} = \sqrt{\frac{SD^2_{\text{true}} + SD^2_{\text{false}}}{2}} \quad (2)$$

The vast majority of experiments (294 out of 303 effects) in our meta-analysis manipulated news veracity within participants, that is, having participants rate both false and true news. Following the Cochrane manual, we accounted for the dependency between ratings that this design generates when calculating the standard error for Cohen's *d*. Precisely, we calculated the standard error for within-participant designs as

$$\text{SE}_{\text{Cohen's } d(\text{within})} = \sqrt{\frac{2(1 - r_{\text{true,false}})}{n} + \frac{\text{Cohen's } d^2}{2n}} \quad (3)$$

where *r* is the correlation between true and false news. Ideally, for each effect size (that is, the meta-analytic units of observation) in our data, we need the estimate of *r*. However, this correlation is generally not reported in the original papers. We could only obtain it for a subset of samples for which we computed the summary statistics ourselves based on the raw data. On the basis of this subset of correlations, we calculated an average correlation, which we then imputed for all effect size calculations. This approach is in line with the Cochrane recommendations for crossover trials[32]. In our case, this average correlation was 0.26.

For the 9 (out of 303) effects from studies that used a between-participant design, we calculated the standard error as

$$\text{SE}_{\text{Cohen's}d\text{(between)}} = \sqrt{\frac{n_{\text{true}} + n_{\text{false}}}{n_{\text{true}}n_{\text{false}}} + \frac{\text{Cohen's}d^2}{2(n_{\text{true}} + n_{\text{false}})}} \qquad (4)$$

For all effect size calculations, we defined the sample size *n* as the number of instances of news ratings. That is, we multiplied the number of participants by the number of news items rated per participant.

**Models.** In our models for the meta-analysis, each effect size was weighted by the inverse of its standard error, thereby giving more weight to studies with larger sample sizes. We used random effects models, which assume that there is not only one true effect size but a distribution of true effect sizes[122]. These models assume that variation in effect sizes is not due to sampling error alone, and thereby allow modelling other sources of variance. We estimated the overall effect of our outcome variables using a three-level meta-analytic model with random effects at the sample and the publication level. This approach allowed us to account for the hierarchical structure of our data, in which samples (level three) contributed multiple effects (level two), level one being the participant level of the original studies (see ref. 122). A common case where a sample provided several effect sizes occurred when participants rated both politically concordant and discordant news. In this case, if possible, we entered summary statistics separately for the concordant and discordant items, yielding two effect sizes (that is, two different rows in our data frame). Another case where multiple effects per sample occurred was when follow-up studies were conducted on the same participants (but different news items). While our multilevel models account for this hierarchical structure of the data, they do not account for dependencies in sampling error. When one same sample contributes several effect sizes, one should expect their respective sampling errors to be correlated[122]. To account for dependency in sampling errors, we computed cluster-robust standard errors, confidence intervals and statistical tests for all meta-analytic estimates.

To assess the effect of moderator variables, we used meta-regressions. We calculated a separate regression for each moderator, by adding the moderator variable as a fixed effect to the multilevel meta-models presented above. We pre-registered a list of six moderator variables to test. These included the 'country' of participants (levels: United States vs all other countries), 'political concordance' (levels: politically concordant vs politically discordant), 'news family' (levels: political, including both concordant and discordant vs covid related vs other, including categories as diverse as history, environment, health, science and military-related news items), the 'format' in which the news were presented (levels: headline only vs headline and picture vs headline, picture and lede), whether news items were accompanied by a 'source' or not, and the 'response scale' used (levels: 4-point vs binary vs 6-point vs 7-point vs other, for all other numeric scales that were not frequent). We ran an additional regression for two non-pre-registered variables, namely, the 'symmetry of scales' (levels: perfectly symmetrical vs imperfectly symmetrical) and 'false news selection' (levels: taken from fact-checking sites vs verified by researchers). We further descriptively checked whether the 'proportion of true news' among all news would yield differences.

**Publication bias.** We ran standard procedures for detecting publication bias. However, we did not a priori expect publication bias to be present because our variables of interest were not those of interest to the researchers of the original studies: researchers generally set out to test factors that alter discernment, and not the state of discernment in the control group. No study measured skepticism bias in the way we define it here.

Regarding discernment, we found evidence that smaller studies tended to report larger effect sizes, according to Egger's regression test (Extended Data Fig. 2; see also Supplementary Section E). We did not find evidence for asymmetry regarding skepticism bias. However, it is unclear how meaningful these results are. As illustrated by the funnel plot, there is generally high between-effect size heterogeneity: even when focusing only on the most precise effect sizes (top of the funnel), the estimates vary substantially. It thus seems reasonable to assume that most of the dispersion of effect sizes does not arise from the studies' sampling error, but from the studies estimating different true effects. Further, even the small studies are relatively high powered, suggesting that they would have yielded significant, publishable results even with smaller effect sizes. Lastly, Egger's regression test can lead to an inflation of false positive results when applied to standardized mean differences[122,123].

We did not find any evidence to suspect *P*-hacking for either discernment or skepticism bias from visually inspecting *P*-curves for both outcomes (Extended Data Fig. 3).

**Reporting summary**

Further information on research design is available in the Nature Portfolio Reporting Summary linked to this article.

## Data availability

The extracted data used to produce our results are available on OSF at https://doi.org/10.17605/OSF.IO/96ZBP (ref. 124).

## Code availability

The code used to create all results (including tables and figures) of this manuscript is also available on OSF at https://doi.org/10.17605/OSF.IO/96ZBP (ref. 124).

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

## Acknowledgements

We thank A. Allard, H. Mercier, G. Pennycook, A. Modirrousta-Galian and B. Tappin for their valuable feedback on earlier versions of the manuscript. J.P. received funding from SCALUP ANR grant ANR-21-CE28-0016-01. S.A. received funding from the European Research Council (ERC) under the European Union's Horizon 2020 research and innovation programme (grant agreement no. 883121). The funders had no role in study design, data collection and analysis, decision to publish or preparation of the manuscript.

## Author contributions

J.P. conceptualized the project, performed systematic literature search, developed methodology, acquired software, conducted formal analysis, data curation and visualization, wrote the original draft, and reviewed and edited the manuscript. S.A. conceptualized the project, performed systematic literature search, wrote the original draft, and reviewed and edited the manuscript.

## Funding

## Competing interests

The authors declare no competing interests.

## Additional information

**Extended data** is available for this paper at

**Supplementary information** The online version
contains supplementary material available at

**Correspondence and requests for materials** should be addressed to
Sacha Altay.

**Peer review information** *Nature Human Behaviour* thanks Chang Lu,
Jonathan van 't Riet and the other, anonymous, reviewer(s) for their
contribution to the peer review of this work. Peer reviewer reports are
available.

**Publisher's note** Springer Nature remains neutral with regard to
jurisdictional claims in published maps and institutional affiliations.

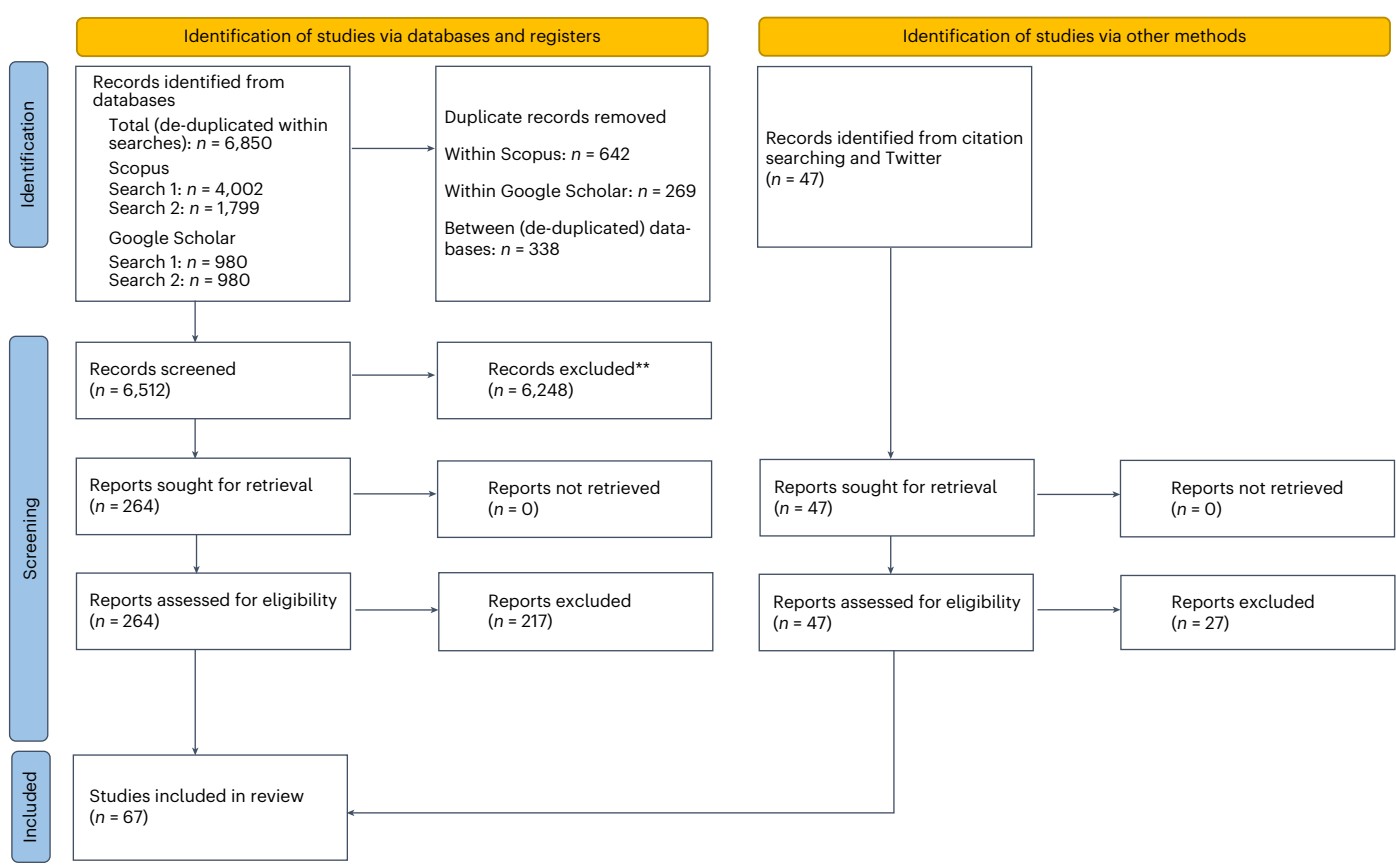

**Extended Data Fig. 1 | PRISMA flow diagram.** A flow diagram for the systematic literature review, based on the 2020 PRISMA template.

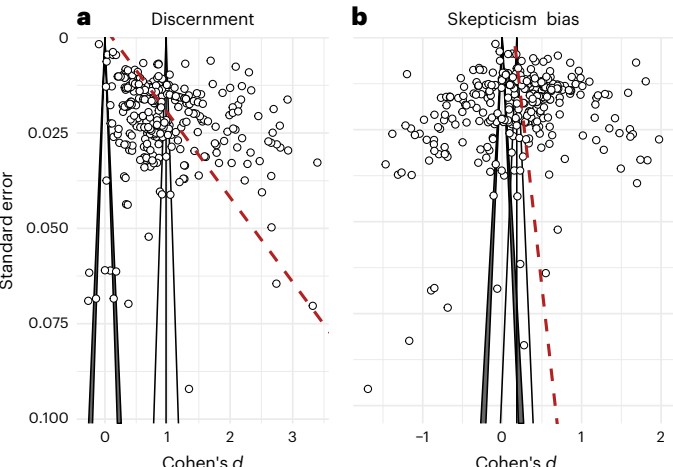

**Extended Data Fig. 2 | Funnel plots for discernment and skepticism bias.** Dots represent effect sizes for both discernment (**a**) and skepticism bias (**b**). In the absence of publication bias and heterogeneity, one would then expect to see the points forming a funnel shape, with the majority of the points falling inside of the pseudo-confidence region centred around the average effect (solid vertical line) estimate, with contours corresponding to the 95% confidence interval, i.e. ±1.96 s.e. (the standard error value from the *y* axis). The second funnel represents the expected shape under the Null-hypothesis, with the shaded area marking statistical significance between the 5% and the 1% level, and everything outside the contours corresponding to the 1% level. The dashed red regression line illustrates the estimate of the Egger's regression test. For discernment (but not for skepticism bias), the slope differs significantly from zero (Supplementary Section E).

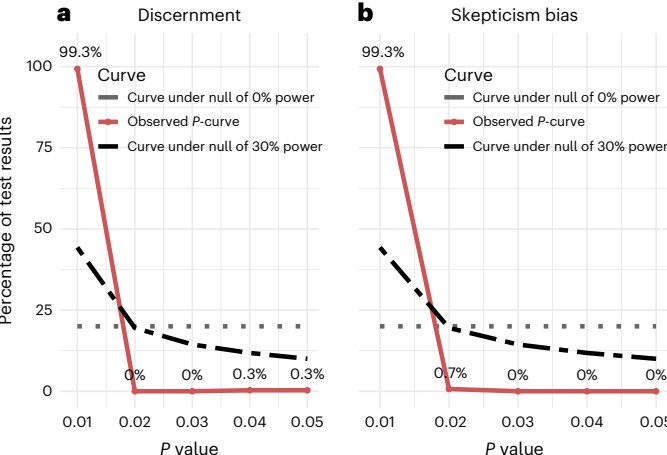

**Extended Data Fig. 3 | *P*-curves for discernment and skepticism bias. a,b,** The *P*-curves show the percentage of effect sizes for a given *P* value within the range 0.01–0.05. All values smaller than 0.01 are rounded off to that value. The dashed/dotted reference lines indicate the expected percentage of studies for a given *P* value, assuming that there is a true effect and certain statistical power to detect it (either 0%, grey dotted line, or 30%, black dashed line, power). The observed *P*-curve is negatively sloped and heavily right skewed (the tail points to the right) for both outcomes, which suggests no evidence of *P*-hacking.

# Reporting Summary

## Statistics

For all statistical analyses, confirm that the following items are present in the figure legend, table legend, main text, or Methods section.

| n/a | Confirmed | |
|---|---|---|
| ☐ | ☒ | The exact sample size (*n*) for each experimental group/condition, given as a discrete number and unit of measurement |
| ☐ | ☒ | A statement on whether measurements were taken from distinct samples or whether the same sample was measured repeatedly |
| ☐ | ☒ | The statistical test(s) used AND whether they are one- or two-sided<br>*Only common tests should be described solely by name; describe more complex techniques in the Methods section.* |
| ☐ | ☒ | A description of all covariates tested |
| ☐ | ☒ | A description of any assumptions or corrections, such as tests of normality and adjustment for multiple comparisons |
| ☐ | ☒ | A full description of the statistical parameters including central tendency (e.g. means) or other basic estimates (e.g. regression coefficient) AND variation (e.g. standard deviation) or associated estimates of uncertainty (e.g. confidence intervals) |
| ☐ | ☒ | For null hypothesis testing, the test statistic (e.g. *F*, *t*, *r*) with confidence intervals, effect sizes, degrees of freedom and *P* value noted<br>*Give P values as exact values whenever suitable.* |
| ☒ | ☐ | For Bayesian analysis, information on the choice of priors and Markov chain Monte Carlo settings |
| ☐ | ☒ | For hierarchical and complex designs, identification of the appropriate level for tests and full reporting of outcomes |
| ☐ | ☒ | Estimates of effect sizes (e.g. Cohen's *d*, Pearson's *r*), indicating how they were calculated |

*Our web collection on statistics for biologists contains articles on many of the points above.*

## Software and code

Policy information about availability of computer code

| | |
|---|---|
| Data collection | We used R version 4.2.2 (2022-10-31), Rstudio version 2024.9.0.375, and the tidyverse package version 2.0.068 for extracting data from papers.The extracted data used to produce our results are available on the OSF project page (https://osf.io/96zbp/). |
| Data analysis | We conducted all analyses in R version 4.2.2 (2022-10-31) using Rstudio version 2024.9.0.375 and the tidyverse package version 2.0.068. For effect size calculations, we rely on the escalc(), for models on therma.mv(), for clustered standard errors on the robust() function, all from the metafor package version 4.6.069. The code used to create all results (including tables and figures) of this manuscript is also available on the OSF project page (https://osf.io/96zbp/). |

For manuscripts utilizing custom algorithms or software that are central to the research but not yet described in published literature, software must be made available to editors and reviewers. We strongly encourage code deposition in a community repository (e.g. GitHub). See the Nature Portfolio guidelines for submitting code & software for further information.

## Data

Policy information about availability of data

All manuscripts must include a data availability statement. This statement should provide the following information, where applicable:
- Accession codes, unique identifiers, or web links for publicly available datasets
- A description of any restrictions on data availability
- For clinical datasets or third party data, please ensure that the statement adheres to our policy

The extracted data used to produce our results are available on the OSF project page (https://osf.io/96zbp/). For our literature search, we relied on the Scopus database (https://www.scopus.com/search/form.uri?display=basic#basic).

## Research involving human participants, their data, or biological material

Policy information about studies with human participants or human data. See also policy information about sex, gender (identity/presentation), and sexual orientation and race, ethnicity and racism.

| | |
|---|---|
| Reporting on sex and gender | No reporting on sex or gender. |
| Reporting on race, ethnicity, or other socially relevant groupings | No reporting on race, ethnicity, or other socially relevant groupings. |
| Population characteristics | No population covariates were measured, except for country. |
| Recruitment | Most studies included in our meta-analysis recruited participants online, some of which used representative samples. |
| Ethics oversight | None. |

Note that full information on the approval of the study protocol must also be provided in the manuscript.

## Field-specific reporting

Please select the one below that is the best fit for your research. If you are not sure, read the appropriate sections before making your selection.

☐ Life sciences  ☒ Behavioural & social sciences  ☐ Ecological, evolutionary & environmental sciences

For a reference copy of the document with all sections, see nature.com/documents/nr-reporting-summary-flat.pdf

## Behavioural & social sciences study design

All studies must disclose on these points even when the disclosure is negative.

| | |
|---|---|
| Study description | We conducted a meta-analysis, i.e. took a purely quantitative approach in our systematic review. |
| Research sample | We selected all our studies systematically, following the PRISMA guidelines. Most samples of included studies were recruited online, via platforms such as Amazon MTurk or Prolific. Some studies recruited representative samples of the respective country population. |
| Sampling strategy | We selected all our studies systematically, following the PRISMA guidelines. Our inclusion criteria are documented in the article. |
| Data collection | We retrieved the relevant summary statistics directly from the paper for 21 papers, calculated them ourselves based on publicly available raw data for 31 papers, and got them from the authors after request for 15 papers. For the second systematic search, both authors screened search results independently. In case of conflicting decisions, an article passed onto the next stage (i.e. receive abstract screening or full text assessment). |
| Timing | We conducted our first search on March 2, 2023. After submitting a manuscript version, reviewers remarked that not including the terms "misinformation" or "disinformation" in our search string might have omitted relevant results. On March 22nd, 2024, we therefor conducted a second, pre-registered (https://osf.io/yn6r2) search using an extended query string. |
| Data exclusions | We document all exclusion phases following the PRISMA protocol in our article. Specifically, we stored 264 records in the reference management system Zotero for retrieval. Of those, we rejected a total of 217 papers that did not meet our inclusion criteria. We rejected 87 papers based on their abstract and 130 after assessment of the full text. All exclusion decisions are documented in a publicly available spreadsheet that can be found on the OSF project page (https://osf.io/96zbp/). |
| Non-participation | We rely on published data and that information was not available in most cases. |
| Randomization | Almost all studies used a within-participant design. Between-participant studies randomized participants to see either true or false |

# Reporting for specific materials, systems and methods

We require information from authors about some types of materials, experimental systems and methods used in many studies. Here, indicate whether each material, system or method listed is relevant to your study. If you are not sure if a list item applies to your research, read the appropriate section before selecting a response.

## Materials & experimental systems

| n/a | Involved in the study |
|-----|----------------------|
| ☒ | ☐ Antibodies |
| ☒ | ☐ Eukaryotic cell lines |
| ☒ | ☐ Palaeontology and archaeology |
| ☒ | ☐ Animals and other organisms |
| ☒ | ☐ Clinical data |
| ☒ | ☐ Dual use research of concern |
| ☒ | ☐ Plants |

## Methods

| n/a | Involved in the study |
|-----|----------------------|
| ☒ | ☐ ChIP-seq |
| ☒ | ☐ Flow cytometry |
| ☒ | ☐ MRI-based neuroimaging |

## Plants

Seed stocks

*Report on the source of all seed stocks or other plant material used. If applicable, state the seed stock centre and catalogue number. If plant specimens were collected from the field, describe the collection location, date and sampling procedures.*

Novel plant genotypes

*Describe the methods by which all novel plant genotypes were produced. This includes those generated by transgenic approaches, gene editing, chemical/radiation-based mutagenesis and hybridization. For transgenic lines, describe the transformation method, the number of independent lines analyzed and the generation upon which experiments were performed. For gene-edited lines, describe the editor used, the endogenous sequence targeted for editing, the targeting guide RNA sequence (if applicable) and how the editor was applied.*

Authentication

*Describe any authentication procedures for each seed stock used or novel genotype generated. Describe any experiments used to assess the effect of a mutation and, where applicable, how potential secondary effects (e.g. second site T-DNA insertions, mosiacism, off-target gene editing) were examined.*

