## [Peer Review File · Nature Human Behaviour]

Spotting false news and doubting true news: a systematic review and meta-analysis on news judgments

Corresponding Author: Dr Sacha Altay

Version 0:

Decision Letter:

20th February 2024

Dear Dr Altay,

Thank you once again for your manuscript, entitled "Spotting False News and Doubting True News: A Meta-Analysis of News Judgements", and for your patience during the peer review process.

Your Article has now been evaluated by 3 referees. You will see from their comments copied below that, although they find your work of potential interest, they have raised quite substantial concerns. In light of these comments, we cannot accept the manuscript for publication, but would be interested in considering a revised version if you are willing and able to fully address reviewer and editorial concerns.

We hope you will find the referees' comments useful as you decide how to proceed. If you wish to submit a substantially revised manuscript, please bear in mind that we will be reluctant to approach the referees again in the absence of major revisions. We are committed to providing a fair and constructive peer-review process. Do not hesitate to contact us if there are specific requests from the reviewers that you believe are technically impossible or unlikely to yield a meaningful outcome.

To guide the scope of the revisions, the editors discuss the referee reports in detail within the team, including with the chief editor, with a view to (1) identifying key priorities that should be addressed in revision and (2) overruling referee requests that are deemed beyond the scope of the current study. We hope that you will find the prioritised set of referee points to be useful when revising your study. Please do not hesitate to get in touch if you would like to discuss these issues further.

In particular, your revision must address the following (as well as all other reviewer comments):

(1) Reviewers 1 and 3 raise important concerns about the underlying literature. Specifically, they are concerned about individuals' ability of telling apart true and fake news, the ecological validity of the experimental settings and the generalisability of the findings beyond the COVID-19 context. Please carefully revise your manuscript in response to these concerns, ensuring you discuss the findings in a nuanced manner and avoid overinterpretation of your findings."

(2) Reviewer 2 is concerned that your search string excludes a substantial portion of relevant literature, as false/fake news intersect with misinformation/disinformation. We share this concern and ask that you rerun your search, using an expanded search string, in order to ensure that all of the relevant literature is identified.

If you wish to submit a suitably revised manuscript, we would hope to receive it within 6 months. I would be grateful if you could contact us as soon as possible if you foresee difficulties with meeting this target resubmission date.

- Include a "Response to the editors and reviewers" document detailing, point-by-point, how you addressed each editor and referee comment. If no action was taken to address a point, you must provide a compelling argument. When formatting this document, please respond to each reviewer comment individually, including the full text of the reviewer comment verbatim followed by your response to the individual point. This response will be used by the editors to evaluate your revision and sent back to the reviewers along with the revised manuscript.
- Highlight all changes made to your manuscript or provide us with a version that tracks changes.

Link Redacted

Thank you for the opportunity to review your work. Please do not hesitate to contact me if you have any questions or would like to discuss the required revisions further.

[redacted]

Reviewer expertise:

Reviewer #1: political psychology, mass media, public opinion

Reviewer #2: fake news discernment, meta-analysis

Reviewer #3: political communication, meta-analysis

REVIEWER COMMENTS:

Reviewer #1:

Remarks to the Author:

The study is generally well done and will be of interest to many researchers. Since this is a meta analysis, I don't have much to say about the methods. My concerns lie with the body of literature overall.

People can only spot truth reliably if they know something about the claim in question or have appropriate priors for doing so. Thus, the idea that people can spot truth upon exposure to a claim is kind of ridiculous. If we could see truth just by exposure to a claim, why did I just read your paper? Shouldn't I have known your paper was true (or false) just by reading the abstract? Why are we doing studies at all then - why can't we just put forward hypotheses and tell just by looking at them which ones are true and which are not? The answer is that we cannot, and that finding truth is hard. Truth is not self-evident, and certainly not self-evident to all. This is why scientists have jobs.

The study's ultimate conclusion is that crowdsourcing factchecks can work because people can see truth. But consider this: a large group of people might be able to guess the number of marbles in a jar upon seeing the size of the jar, the size of the marbles, and the fullness of the jar. But what happens if people could not see the size of the jar, the marbles, or the fullness of the jar? The guesses would be all over the place, and likely not average out to a close estimate. Thus, people's ability to spot truth is very much a function of the specific claims examined and what the guessers know about the claims in advance and what their priors are.

This is where the claims being judged as true and false themselves start to matter so much. The studies examined here seem to focus on only a few topics, one of the biggest being COVID. But if you move away from hot topics, there are millions of things that people know nothing about and could not possibly be able to guess the truth about reliably. I think this paper needs more discussion about this. Lots of people believe wrong things, so how then are people able to just see truth? Are the people who come to the wrong conclusion lying or insane? Further, the factors that are associated with guessing the truth of a claim correctly likely vary as well from claim to claim.

To sum then, my concern is that the results of all of the studies are dependent on the headlines/claims being proffered and the priors of the people doing the guessing. So, I think this severely limits the generalizability of all of the papers included in this meta-analysis. So, I would like to see the authors address the epistemology issue and the generalizability issue.

Reviewer #2:

Remarks to the Author:

This manuscript has made a valuable contribution to the study of false news. It provides clear guidance on how to improve the public's ability to deal with false news. It is very commendable that the entire article is methodologically rigorous, with a clear description and in-depth discussion. We believe that this meta-analysis will provide valuable clues for future research on false news. Here are some minor suggestions for this manuscript:

1. The authors propose an interesting concept, skepticism bias. This certainly adds to the current assessment of false news. I suggest that the authors give the reference to the calculation method. If this calculation method is proposed in this paper, a full methodological justification needs to be provided. As well, references of the calculation method of Discernment should be provided (Pennycook & Rand, 2019; Roozenbeek et al., 2022).

2. I have some concerns about the authors' search strategy, in which the authors only searched for "false news" OR "fake news" OR "false stor*". However, fake news is often a hybrid with the concepts of misinformation and disinformation (Scheufele & Krause, 2019; Zhou & Zafarani, 2020). This will lead to some literature being left out (e.g. Maertens et al., 2021; Roozenbeek et al., 2022; Hu et al., 2023).

3. Authors need to provide the start date and the deadline for searching the literature.

4. I believe that suggestions for methodologies of measuring and assessing false news can be added to the section of the discussion, which would provide more specific guidance for future research.

5. The definition of perfect symmetrical labels and imperfectly symmetrical labels confused me. (Method-Data-Deviations from eligibility criteria AND Appendix B). I wish the authors could provide a more clear description

Reference

- Hu, B., Ju, X.-D., Liu, H.-H., Wu, H.-Q., Bi, C., & Lu, C. (2023). Game-based inoculation versus graphic-based inoculation to combat misinformation: A randomized controlled trial. *Cognitive Research: Principles and Implications*, 8(1), 49. <https://doi.org/10.1186/s41235-023-00505-x>
- Maertens, R., Roozenbeek, J., Basol, M., & van der Linden, S. (2021). Long-term effectiveness of inoculation against misinformation: Three longitudinal experiments. *Journal of Experimental Psychology: Applied*, 27(1), 1. <https://doi.org/10.1037/xap0000315>
- Pennycook, G., & Rand, D. G. (2019). Lazy, not biased: Susceptibility to partisan fake news is better explained by lack of reasoning than by motivated reasoning. *Cognition*, 188, 39–50. <https://doi.org/10.1016/j.cognition.2018.06.011>
- Roozenbeek, J., van der Linden, S., Goldberg, B., Rathje, S., & Lewandowsky, S. (2022). Psychological inoculation improves resilience against misinformation on social media. *Science Advances*, 8(34), eabo6254. <https://doi.org/10.1126/sciadv.abo6254>
- Scheufele, D. A., & Krause, N. M. (2019). Science audiences, misinformation, and fake news. *Proceedings of the National Academy of Sciences of the United States of America*, 116(16), 7662–7669. <https://doi.org/10.1073/pnas.1805871115>
- Zhou, X., & Zafarani, R. (2020). A Survey of Fake News: Fundamental Theories, Detection Methods, and Opportunities. *ACM Computing Surveys*, 53(5), 1–40. <https://doi.org/10.1145/3395046>

Reviewer #3:

Remarks to the Author:

The paper under review investigates news consumers' ability to distinguish between false and true headlines by means of a meta-analysis. I would like to commend the authors for their efforts: the paper covers an important topic, uses sound methodology, including preregistration, and is well written. The results are highly informative: when presented with true and false headlines in experimental studies, participants on average rate true headlines as more accurate than false headlines, thus: people seem able to distinguish between true and false headlines.

Any reservations that I have center on my sense that the implications of these results are not as straightforward as the authors suggest. I'm not saying that the authors misrepresent their results – I understand the desire to present a relatively clear story and to not overcomplicate things. I do think, however, that things are more complicated than the authors let on, and that the paper would benefit from acknowledging this. Or at least I would be very interesting in hearing the authors thoughts on this. So with the greatest respect, I have detailed my concerns below, starting with the interpretation of the discernment effect, following up with the interpretation of the skepticism bias effect, and closing with some minor comments and questions.

Implications of the discernment effect

- "most people do not lack the skills to spot false news" (p12) seems a valid conclusion from the present work, and, from the standpoint of truth and democracy, a reassuring one. But this conclusion should perhaps be qualified by "And under the right circumstances they will apply these skills". It seems the included experiments went out of their way to create ideal circumstances for discernment: participants were paid (I assume) to judge headlines, with little distraction, and were instructed to make an explicit veracity judgment. The authors do discuss the importance of motivation in the Discussion section (p12) and of prompting participants to consider veracity (p13), but I think it's worthwhile to consider the more generally artificial nature of many of the included experiments explicitly. The discernment effect may be a good indication of people's skills to spot false news, but as an indication of how likely people are to actually spot false news in real life, it is probably a rather large overestimate. I'm aware that the authors do not interpret it as such, but I worry that many readers will. The authors already consider that real-life news consumption does not come with prompts to think of accuracy (p13), but they may also want to consider that real life news consumption depends heavily on self-selection and algorithmic selection (not forced exposure), and that both false and true headlines can be forwarded and commented on by one's social network (rather than be presented without any social cues).

- The ability to spot false news surely depends on the specific news. It would be easy to devise an experiment with a selection of true and false headlines on esoteric subjects where participants will certainly lack the required knowledge to spot the difference. It would be similarly easy to devise an experiment that is all but certain to find a large difference. I'm sure most of this work is done in good faith, but don't we know enough about (unconscious) researcher bias to consider this aspect of the experiments? I'm reassured to see that some studies used a random selection of true headlines (Appendix G), but the selection of false headlines remains a bit of black box. The implication seems to be: "given the selection of headlines that were chosen by researchers in this field, most people do not lack the skills to spot false news"

Skepticism bias

- The same issues of ecological validity also apply to skepticism. Prompting participants to rate the veracity of headlines alerts them to the possibility of fake headlines and may make them more skeptical than they would otherwise be. The problem here is greater, because here the authors do interpret the result as a reflection of what news consumers would do in real life. I think the paper would benefit from an acknowledgment of the fact that the skepticism bias may have been inflated by the experimental context. I do appreciate the discussion of the danger posed by excessive skepticism (p12) and by interventions that reduce trust in false and true news alike (p13). But I also feel that the skepticism bias effect found here should not be overstated.

- Calculating the differences between the mean scores and the end-point of the scales seems like an acceptable way of assessing

skepticism bias, but it does assume that only veracity judgments are assessed, and not, for instance, certainty. But perhaps participants held two thoughts in their head: how accurate they thought the headline was and how certain they were about this. Even for headlines that they accept as true, participants may have been reluctant to indicate the highest score on the veracity scale. After all, can you ever be 100% sure that something is true? Choosing the most extreme score (the lowest score) is easier in the case of headlines that participants think are obvious nonsense. I'm not sure that this indicates a worrying rejection of true headlines. Rather, it may be a simple acknowledgment that some things are definitively untrue, whereas there are not a lot of things that you can be certain are definitively and irrevocably true. Could this also be the reason why studies using 7-point scales have such lower skepticism bias? Because participants had more opportunity there to indicate the veracity of true headlines while shunning the most extreme answer? I realize this is not the only possible explanation of this moderator effect, but I do feel the moderation by the type of accuracy scale (p9) raises questions about the employed methodology. I wonder what would happen if the analysis would be run on the average percentage of true headlines that participants rate as true (higher than the midpoint of the scale) minus the average percentage of false headlines that participants rate as false. Perhaps I misunderstand, but I think this would be similar to the studies using binary scales, and for those studies the effect is also much smaller (p9). Is skepticism bias a direct result of the 4-point scale?

Minor issues

- Crowdsourced fact-checking initiatives are mentioned rather prominently in the Discussion as well as the Abstract, but to me that feels like a rather big leap. I would personally prefer a more in-depth discussion of what these results mean (see above).
- On page 4, the authors say that they only included studies with neutral control conditions. Does this mean that only the data from the control conditions was used? I think examples would be nice of the kinds of studies that were included, so that readers who are not familiar with this literature get a sense of the experiments.
- On a similar note, when did the same sample contribute several discernment effect sizes (p7)? Is this for instance when politically discordant and concordant groups were used and different effect sizes were included? Or is the explanation the same as for the skepticism effect size (p8).
- For the studies using a binary scale, am I correct in inferring that this basically comes down to calculating the average proportion of times participants correctly identified a false headline minus the average proportion of times participants correctly identified a true headline?
- Are the difference scores reported in the Moderators section meta-regression weights? Why not (also) report the composite effect sizes for the subgroups?
- The interpretation of the effect of political concordance seems off to me (p13). The authors state that participants became more skeptical when the headlines were discordant, but the opposite is also true: they became more gullible when the headlines were concordant. Suggesting that participants became "excessively skeptical" for discordant but not "excessively gullible" for concordant headlines seems slightly misleading to me. That's simply a result of the starting position being a slight bias toward skepticism. A bias for which the interpretation is not entirely unproblematic, I might add (see above).

Version 1:

Decision Letter:

Our ref: NATHUMBEHAV-23103308A

23rd July 2024

Dear Dr. Altay,

Thank you for submitting your revised manuscript "Spotting False News and Doubting True News: A Meta-Analysis of News Judgements" (NATHUMBEHAV-23103308A). It has now been seen by the original referees and their comments are below. As you can see, the reviewers find that the paper has improved in revision. We will therefore be happy in principle to publish it in Nature Human Behaviour, pending minor revisions to satisfy the referees' final requests and to comply with our editorial and formatting guidelines.

We are now performing detailed checks on your paper and will send you a checklist detailing our editorial and formatting requirements in the following weeks. Please do not upload the final materials and make any revisions until you receive this additional information from us.

[redacted]

Reviewer #1 (Remarks to the Author):

The authors have addressed my comments thoroughly. I support publication.

Reviewer #2 (Remarks to the Author):

Dear Author:

Thank you for your thoughtful response to my initial review of your manuscript. I am pleased to see that you have addressed my concerns and made substantial revisions to the manuscript. I believe that the revised manuscript is significantly improved and makes a valuable contribution to the study of misinformation. I recommend that it be accepted for publication.

Reviewer #3 (Remarks to the Author):

Dear Editor,

The authors have done a commendable job addressing the questions raised in my review. The manuscript was always going to constitute a fine contribution to the field; I hope my review has challenged the authors a little bit to (re)consider their results some more. Although I myself would perhaps make different choices in places - for instance the crowd-sourced fact checking discussion - the authors are obviously more knowledgeable in this area than I am and have spent far more time thinking about these issues.

The issue of skepticism bias and concordance perhaps offers interesting avenues for future research, but I completely understand the authors choice to not address it in the Discussion.

Version 2:

Decision Letter:

Dear Dr Altay,

We are pleased to inform you that your Article "Spotting false news and doubting true news: a systematic review and meta-analysis on news judgments", has now been accepted for publication in Nature Human Behaviour.

Please note that *Nature Human Behaviour* is a Transformative Journal (TJ). Authors may publish their research with us through the traditional subscription access route or make their paper immediately open access through payment of an article-processing charge (APC). Authors will not be required to make a final decision about access to their article until it has been accepted. [Find out more about Transformative Journals](https://www.springernature.com/gp/open-research/transformative-journals)

You can now use a single sign-on for all your accounts, view the status of all your manuscript submissions and reviews, access

usage statistics for your published articles and download a record of your refereeing activity for the Nature journals.

[redacted]

P.S. Click on the following link if you would like to recommend Nature Human Behaviour to your librarian
<http://www.nature.com/subscriptions/recommend.html#forms>

** Visit the Springer Nature Editorial and Publishing website at http://editorial-jobs.springernature.com?utm_source=ejp_NHumB_email&utm_medium=ejp_NHumB_email&utm_campaign=ejp_NHumB for more information about our career opportunities. If you have any questions please click [here](mailto:editorial.publishing.jobs@springernature.com).

Open Access This Peer Review File is licensed under a Creative Commons Attribution 4.0 International License, which permits use, sharing, adaptation, distribution and reproduction in any medium or format, as long as you give appropriate credit to the original author(s) and the source, provide a link to the Creative Commons license, and indicate if changes were made. In cases where reviewers are anonymous, credit should be given to 'Anonymous Referee' and the source. The images or other third party material in this Peer Review File are included in the article's Creative Commons license, unless indicated otherwise in a credit line to the material. If material is not included in the article's Creative Commons license and your intended use is not permitted by statutory regulation or exceeds the permitted use, you will need to obtain permission directly from the copyright holder.

Reviewer 1:

The study is generally well done and will be of interest to many researchers. Since this is a meta analysis, I don't have much to say about the methods. My concerns lie with the body of literature overall.

People can only spot truth reliably if they know something about the claim in question or have appropriate priors for doing so. Thus, the idea that people can spot truth upon exposure to a claim is kind of ridiculous. If we could see truth just by exposure to a claim, why did I just read your paper? Shouldn't I have known your paper was true (or false) just by reading the abstract? Why are we doing studies at all then - why can't we just put forward hypotheses and tell just by looking at them which ones are true and which are not? The answer is that we cannot, and that finding truth is hard. Truth is not self-evident, and certainly not self-evident to all. This is why scientists have jobs.

The study's ultimate conclusion is that crowdsourcing factchecks can work because people can see truth. But consider this: a large group of people might be able to guess the number of marbles in a jar upon seeing the size of the jar, the size of the marbles, and the fullness of the jar. But what happens if people could not see the size of the jar, the marbles, or the fullness of the jar? The guesses would be all over the place, and likely not average out to a close estimate. Thus, people's ability to spot truth is very much a function of the specific claims examined and what the guessers know about the claims in advance and what their priors are.

Authors:

We very much agree that people cannot spot truth upon exposure to any claim and that people need to have some prior accurate knowledge about the world to do so. For instance, past work has shown that people who follow the news more and are more interested in the news (or politics) are better at discerning true from false news. Thus, if participants were able to discern true from false news in the meta-analysis, it is likely in part because they were somewhat informed about the events depicted in the headlines. Obviously, context matters. For example, Italians would struggle to evaluate the veracity of claims about Chilean politics, and vice versa. Yet, in the studies included in the meta-analysis, participants evaluated culturally relevant claims, such as Italians evaluating claims about Italian politics. Moreover, we are not claiming that people can spot truth upon exposure to any claim, but that they can do so upon exposure to true news and fact-checked false news that are culturally relevant. We are now clearer in the manuscript about this:

“Our results do not speak to the reasons why participants were able to discern true from false news. Participants were generally asked to rate culturally relevant news stories, such as Brazilians rating Brazilian news stories. Thus, participants most likely relied on some prior knowledge to evaluate news veracity. Participants would probably not have been capable of discerning news stories on which they completely lack relevant prior knowledge, e.g. culturally distant news stories.” (p.12)

We are also clearer about the fact that our results do not generalize to misinformation more broadly, and certainly do not generalize to ‘any claim’.

“Third, our results reflect choices made by researchers about news selection. As we lay out in Appendix G, we believe that this selection bias mostly concerns the false news items. The vast majority of studies in our meta-analysis relied on fact-checked false

news, determined by fact-checking websites (e.g. Snopes, PolitiFact). By contrast, three papers^{39,59,60} automated their news selection by scraping headlines from media outlets in real-time, and had both participants and fact-checkers (or the researchers themselves, in the case of⁵⁹) rated the veracity of the headlines shortly after. The three studies (53 effect sizes; 10170 participants; all in the United States) find (i) lower discernment and (ii) a negative skepticism (i.e. a credulity) bias. As we discuss in Appendix G, this is likely because they included false news that are harder to fact-check (and not typically fact-checked) or because the news are less false than the typical fact-checked false news. Yet, more work is needed to investigate whether the skepticism bias documented here is due to the selection of fact-checked false news or to something else. This highlights the importance of news selection in misinformation research: Researchers need to think carefully about what population of news they sample from, and be clear about the generalizability of their findings^{42,61}. Overall, our results are informative about people's ability to spot fact-checked false news, and about their doubts towards mainstream true news. However, our results also suggest that people discern worse for more representative samples of misinformation news. More research designed to overcome news selection bias is needed to provide a solid account of how much worse." (p.14/15)

We would like to highlight that this problem of generalization is not new in misinformation research. The famous 2018 Science paper 'The spread of true and false news online' cited more than 8500 times relied on *fact-checked* true and false news and dedicated a large part of the manuscript/appendix to estimating whether the effect generalizes beyond fact-checked news.

Sinan Aral @sinanaral

Love this diagram by @dkroy! It clarifies that we make strong generalizations to "verified contested false news," reasonable generalizations to "contested false news," and weak generalizations to "false news." The (10%) robustness dataset helps us generalize beyond fact-checkers.

News

Fake news*

False news

Contested false news

Contested false news that's been fact-checked

Contested false news that's been fact-checked by snopes.com, politifact.com, factcheck.org, truthorfiction.com, hoax-slayer.com and urbanlegends.about.com & mentioned on Twitter, plus small sample of rumors spread on Twitter**

*Gallup poll (2018) shows that majority of Americans sometimes or always considers not only false news to be "fake news" but also opinion stated as fact, and true news that goes against political interests

**Diagram not to scale, the world of all news is surely WAY bigger than shown! (Study contrasts spread of contested true news, not shown here for simplicity)

Scope of our Science paper

Scope of strong generalization of our study

Scope of weak generalizations

Scope of weaker generalizations

Scope of some conclusions being drawn from our Science paper

10:29 PM · Mar 15, 2018

Similarly, studies estimating the prevalence of misinformation have either relied on URLs of fact-checked false news (vs. mainstream true news) or have conducted analyses at the domain level (e.g., trustworthy vs untrustworthy news sources). Thus, to a large extent, our meta-analysis suffers from the same problems as the rest of the misinformation literature.

Reviewer 1:

This is where the claims being judged as true and false themselves start to matter so much. The studies examined here seem to focus on only a few topics, one of the biggest being COVID. But if you move away from hot topics, there are millions of things that people know nothing about and could not possibly be able to guess the truth about reliably. I think this paper needs more discussion about this. Lots of people believe wrong things, so how then are people able to just see truth? Are the people who come to the wrong conclusion lying or insane? Further, the factors that are associated with guessing the truth of a claim correctly likely vary as well from claim to claim.

Authors:

We agree that there are a lot of topics on which people know absolutely nothing about and would be incapable of evaluating the veracity of claims on these topics. Trivially, someone not familiar with the video game Zelda cannot evaluate the veracity of claims about Hyrule.

In line with the reviewer's previous comment, we are now clearer that the claims included in the meta-analysis are not a random sample of all possible claims one could make about the world, but of news headlines about topics that researchers and fact-checkers deem important, such as politics, health and sciences (see quote on cultural relevance above).

However, we would like to highlight the diversity and number of headlines and topics included in this meta-analysis. Only 13 articles were about COVID-19, 43 articles were on political news, and 20 on "other" news subjects, a category which regroups all news topics not explicitly labeled as "covid" or "political" by the authors of the respective papers, and which includes news topics reaching from health, cancer and science, to economics, history and military covering news.

One might argue that the context of COVID-19 has influenced people's news evaluations more generally, and they might not be representative of what people would otherwise do, even on non-COVID-19 related news. The articles we included in our meta-analysis cover a time period from 2018 to 2024. 33 Articles have been published during the three years that we believe can be said to have been profoundly marked by COVID-19 (2020, 2021, 2022). However, all other articles should not, or to a significantly lesser extent, have suffered from a potential period-bias.

While most studies we reviewed probably used news on "hot topics" (covid, politics, health), these topics seem to be the ones that fact-checkers and researchers deem most relevant and important for society. Further, even within these high-profile news categories, people do of course make mistakes when judging accuracy - they do not discern perfectly and there is certainly room for improvement. But we believe it is an interesting finding—considering overall low news consumption and fear of misinformation—that, on average, people discern rather well.

Reviewer 1: *To sum then, my concern is that the results of all of the studies are dependent on the headlines/claims being proffered and the priors of the people doing the guessing. So, I think this severely limits the generalizability of all of the papers included in this meta-analysis. So, I would like to see the authors address the epistemology issue and the generalizability issue.*

Authors: Of course, we don't see how people could accurately evaluate the veracity of claims without having some prior knowledge about the world pertaining to these claims in some way. Note that some claims may require very minimal knowledge about the world. For instance, to evaluate the claim that "Ben is dead and alive" one only needs some minimal knowledge that people can't be both dead and alive. And to evaluate the claim that "Ben was born in 1620 and is alive today" one only needs to know that humans can't live 400 years. Some false headlines may require such minimal knowledge, as a few of them are about fictive creatures such as Aliens. Yet, we believe that to evaluate the accuracy of most headlines included in the meta-analysis people needed (much) more than such minimal knowledge. We are now clearer in the manuscript that our results reflect the choice made by researchers in news selection, that we cannot extrapolate them to 'any' claim or any news headlines, and can't even extrapolate them to misinformation in general (see previous citations).

Throughout the manuscript (e.g., in the abstract), we made clear that our conclusions are limited to fact-checked false news. However, the term 'false news' appears ~120 times in the manuscript and Appendix, so we did not repeat 'fact-checked' ~120 times, and only added 'fact-checked' to key passages in the manuscript where we present our results (e.g. the abstract and the discussion).

However, as we have tried to argue above, we believe that fact-checked false news represents a category of misinformation that is of interest to many. Acknowledging this selection bias, our results are based on 2167 unique headlines covering various topics, and are therefore unlikely to be overly influenced by specific headlines.

Reviewer 2:

This manuscript has made a valuable contribution to the study of false news. It provides clear guidance on how to improve the public's ability to deal with false news. It is very commendable that the entire article is methodologically rigorous, with a clear description and in-depth discussion. We believe that this meta-analysis will provide valuable clues for future research on false news. Here are some minor suggestions for this manuscript:

Authors: We thank the reviewer for their appreciation of our work.

Reviewer 2: *1. The authors propose an interesting concept, skepticism bias. This certainly adds to the current assessment of false news. I suggest that the authors give the reference to the calculation method. If this calculation method is proposed in this paper, a full methodological justification needs to be provided. As well, references of the calculation method of Discernment should be provided (Pennycook & Rand, 2019; Roozenbeek et al., 2022).*

Authors: We agree with the reviewer that methodological justifications should be made. As the reviewer highlights, the differential calculation method of discernment is standard in the field, and we made sure to add a reference:

"This differential measure of discernment is common in the literature on misinformation 31" (p.5)

The method we used for skepticism bias is, unfortunately, not established. Conceptually, it is akin to the response bias (typically abbreviated "c") in Signal Detection Theory::

“Note that we cannot use more established measures of discernment or response bias, such as Signal Detection Theory, because we rely on mean ratings and not individual ratings. However, in Appendix H, we show that for the studies we have raw data on, our main findings hold when relying on d' (sensitivity) and c (response bias) in Signal Detection Theory.” (p.6)

The reason we introduced Skepticism bias in the first place is that we used summary data in our main analysis, and Signal Detection Theory was thus not applicable. We tried our best to opt for an intuitive measure that we describe with examples before the results section.

Reviewer 2: *2. I have some concerns about the authors' search strategy, in which the authors only searched for “false news” OR “fake news” OR “false stor*”. However, fake news is often a hybrid with the concepts of misinformation and disinformation (Scheufele & Krause, 2019; Zhou & Zafarani, 2020). This will lead to some literature being left out (e.g. Maertens et al., 2021; Roozenbeek et al., 2022; Hu et al., 2023).*

Authors: This was a lot of work, but it was worth it. We expanded the original search string to include the terms “misinformation” and “disinformation”. After removing duplicates with our previous search – 642 between the initial and the revised Scopus search and 269 between the initial and the revised Google Scholar search – we reviewed an additional 1157 results for Scopus and 711 results for Google Scholar. Although we had to exclude many studies that did not consider true news and focused on mis/disinformation, we were able to identify 14 new articles, yielding 71 additional effect sizes from 89'218 participants. Our results for both discernment and skepticism bias are robust to these additions – although both meta-estimates are slightly smaller now. For 10 of the newly added articles, we were able to extract individual level-data, thereby increasing this subset from 12 studies (24'441 participants; 507'754 observations) to 22 studies (42'074 participants; 813'517 observations). With this new subset, we were able to extend some analyses, for example regarding binary vs. likert scales (Appendix D) or misinformation vs. fact-checked false news (Appendix G).

In the revised methods section, we describe the literature review including the revised search. While we were able to add a lot of studies, we had to exclude the candidate papers suggested by the reviewer for the following reasons:

- Maertens, R., Roozenbeek, J., Basol, M., & Van Der Linden, S. (2021). Long-term effectiveness of inoculation against misinformation: Three longitudinal experiments. *Journal of Experimental Psychology: Applied*, 27(1), 1–16. <https://doi.org/10.1037/xap0000315>
- Roozenbeek, J., van der Linden, S., Goldberg, B., Rathje, S., & Lewandowsky, S. (2022). Psychological inoculation improves resilience against misinformation on social media. *Science Advances*, 8(34), eabo6254. <https://doi.org/10.1126/sciadv.abo6254>

The researchers used made-up news items, whereas we only included studies that used real-world news items. This criterion was perhaps our most important inclusion criterion, as the aim of this study was to investigate people's ability to judge news items that they may encounter or have encountered in their everyday lives, and not made-up news items that researchers created to measure some treatment effects.

- Hu, B., Ju, X.-D., Liu, H.-H., Wu, H.-Q., Bi, C., & Lu, C. (2023). Game-based inoculation versus graphic-based inoculation to combat misinformation: A randomized controlled

The researchers did not provide relevant summary statistics and their data does not allow us to calculate accuracy ratings ourselves (they only provide summarized data for discernment, leaving us unable to calculate skepticism bias).

All exclusion decisions are documented in a “excluded.csv” spreadsheet published on the OSF, along with all other replication materials.

Reviewer 2: 3. *Authors need to provide the start date and the deadline for searching the literature.*

Authors: We added search dates in the revised manuscript:

- “We conducted our first search on March 2, 2023 using scopus (search string: “false news” OR “fake news” OR “false stor*” AND “accuracy” OR “discernment” OR “credibilit*” OR “belief” OR “susceptib*”) and google scholar (search string: “Fake news” | “False news”|“False stor*” “Accuracy” | “Discernment”|“Credibility”|“Belief”|“Suceptib*”, no citations, no patents’).” (p.16)
- “After submitting a manuscript version, reviewers remarked that not including the terms “misinformation” or “disinformation” in our search string might have omitted relevant results. On March 22nd, 2024, we therefor conducted a second, pre-registered (<https://osf.io/yn6r2>) search using an extended query string (search string for both scopus and google scholar: “false news” OR “fake news” OR “false stor*” OR “misinformation” OR “disinformation”) AND (“accuracy” OR “discernment” OR “credibilit*” OR “belief” OR “suceptib*” OR “reliab*” OR “vulnerabi*”; see Appendix J for detailed search string).” (p.17)

Both searches were done on a single day, as we exported all search results into data frames and then worked with these data frames during the screening processes. These data frames, for both searches and along with data frames throughout all screening stages, are publicly available on the OSF.

Reviewer 2: 4. *I believe that suggestions for methodologies of measuring and assessing false news can be added to the section of the discussion, which would provide more specific guidance for future research.*

Authors: We agree and we thank the author for the suggestion. We believe that recent papers using a Signal Detection Theory framework have already made the case that, in addition to discernment, response bias is an important measure when assessing people’s news judgments. We mention this literature now:

- “Interventions aimed at reducing misperceptions should therefore consider increasing the acceptance of true news in addition to reducing the acceptance of false news^{13,46}. At the very least, when testing interventions, researchers should evaluate their effect on both true and false news, not just false news⁴⁷. At best, interventions should use methods that allow to estimate discrimination while accounting for response bias, such as Signal Detection Theory, and make sure that apparent increases in discernment are not due to more conservative response bias^{27,48}. This is all the more important given that

recent evidence suggests that many interventions against misinformation, such as media literacy tips⁴⁹, fact-checking⁵⁰, or educational games aimed at inoculating people against misinformation²⁷, may reduce misperceptions of false news at the expense of true news.” (p.13)

We also tried to stress that it is crucial for the field of misinformation to overcome selection bias, if the aim is to make claims on misinformation more generally, and not only on fact-checked false news:

- “This highlights the importance of news selection in misinformation research: Researchers need to think carefully about what population of news they sample from, and be clear about the generalizability of their findings^{42,62}. Overall, our results are informative about people’s ability to spot fact-checked false news, and about their doubts towards mainstream true news. However, our results also suggest that people discern worse for more representative samples of misinformation news. More research designed to overcome news selection bias is needed to provide a solid account of how much worse.” (p.15)

Reviewer 2: *5. The definition of perfect symmetrical labels and imperfectly symmetrical labels confused me. (Method-Data-Deviations from eligibility criteria AND Appendix B). I wish the authors could provide a more clear description*

Authors: Sorry for this oversight! We are now clearer about this:

- “Skepticism bias can only be (meaningfully) computed on scales using symmetrical labels, i.e., the intensity of the labels to qualify true and false news are equivalent (e.g., “True” vs “False” or “Definitely fake” [1] to “Definitely real” [7]). 69% of effects included in the meta-analysis used scales with perfectly symmetrical labels, while 26% used imperfectly symmetrical scale labels, i.e., the intensity of the labels to qualify true and false news are similar but not equivalent is similar but not equivalent to qualify true and false news (e.g., [1] not at all accurate, [2] not very accurate, [3] somewhat accurate, [4] very accurate; here for instance ‘not all accurate’ is stronger than ‘very accurate’)” (p.6)

Reviewer 3:

The paper under review investigates news consumers’ ability to distinguish between false and true headlines by means of a meta-analysis. I would like to commend the authors for their efforts: the paper covers an important topic, uses sound methodology, including preregistration, and is well written. The results are highly informative: when presented with true and false headlines in experimental studies, participants on average rate true headlines as more accurate than false headlines, thus: people seem able to distinguish between true and false headlines.

Any reservations that I have center on my sense that the implications of these results are not as straightforward as the authors suggest. I’m not saying that the authors misrepresent their results – I understand the desire to present a relatively clear story and to not overcomplicate things. I do think, however, that things are more complicated than the authors let on, and that the paper would benefit from acknowledging this. Or at least I would be very interesting in hearing the authors thoughts on this. So with the greatest respect, I have detailed my concerns below, starting with the interpretation of the discernment effect, following up with the interpretation of the skepticism bias effect, and closing with some minor comments and questions.

Authors: We thank the reviewer for their appreciation, and for emphasizing what we, too, see as the most pressing discussion point regarding the limitations of our meta-analyses: the generalization from fact-checked false news to misinformation. We hope the reviewer agrees that (i) stressing this limit in the discussion of the revised manuscript, (ii) including two new studies (three in total) that used automated news selection and are thus less biased in their sample of misinformation, and (iii) running new analysis on this subset of studies (see Appendix G) made our findings more nuanced.

In our manuscript we speak of ‘fact-checked false news’ now, instead of only ‘false news’. However, the term ‘false news’ appears ~120 times in the manuscript and Appendix, so we did not repeat ‘fact-checked’ ~120 times, and only added ‘fact-checked’ to key passages in the manuscript where we present our results (e.g. the abstract and the discussion). We highlight relevant passages of the discussion below.

Reviewer 3:

Implications of the discernment effect

- “most people do not lack the skills to spot false news” (p12) seems a valid conclusion from the present work, and, from the standpoint of truth and democracy, a reassuring one. But this conclusion should perhaps be qualified by “And under the right circumstances they will apply these skills”. It seems the included experiments went out of their way to create ideal circumstances for discernment: participants were paid (I assume) to judge headlines, with little distraction, and were instructed to make an explicit veracity judgment. The authors do discuss the importance of motivation in the Discussion section (p12) and of prompting participants to consider veracity (p13), but I think it’s worthwhile to consider the more generally artificial nature of many of the included experiments explicitly. The discernment effect may be a good indication of people’s skills to spot false news, but as an indication of how likely people are to actually spot false news in real life, it is probably a rather large overestimate. I’m aware that the authors do not interpret it as such, but I worry that many readers will. The authors already consider that real-life news consumption does not come with prompts to think of accuracy (p13), but they may also want to consider that real life news consumption depends heavily on self-selection and algorithmic selection (not forced exposure), and that both false and true headlines can be forwarded and commented on by one’s social network (rather than be presented without any social cues).

Authors: Of course, we couldn’t agree more. As the reviewer noted, we already mentioned the artificial experimental context in the discussion, but we now state this, we hope, more explicitly:

“First, participants evaluated the news stories in artificial settings that do not mimic the real-world. For instance, the mere fact of asking participants to rate the accuracy of the news stories may have increased discernment by increasing attention to accuracy³³. When browsing on social media, people may be less discerning (and perhaps less skeptical) than in experimental settings because they would pay less attention to accuracy³⁷. However, given people’s low exposure to misinformation online⁵⁷, most people may protect themselves from misinformation not by detecting misinformation on the spot, but by relying on the reputation of the sources and avoiding unreliable sources⁵⁸.” (p.14)

Regarding news selection, we had much room for improvement. We agree that our sample is biased towards fact-checked false news and thus not a random sample of misinformation. We

tried to address this issue by (i) stressing it in the discussion of the revised manuscript, and (ii) including two new studies (three in total) that used automated news selection and are thus less biased in their sample of misinformation (see Appendix G).

In our revised discussion section, we now write:

“Third, our results reflect choices made by researchers about news selection. As we lay out in Appendix G, we believe that this selection bias mostly concerns the false news items. The vast majority of studies in our meta-analysis relied on fact-checked false news, determined by fact-checking websites (e.g. Snopes, PolitiFact). By contrast, three papers^{39,60,61} automated their news selection by scraping headlines from media outlets in real-time, and had both participants and fact-checkers (or the researchers themselves, in the case of⁶⁰) rated the veracity of the headlines shortly after. The three studies (effect sizes; participants; all in the United States) find (i) lower discernment and (ii) a negative skepticism (i.e. a credulity) bias. As we discuss in Appendix G, this is likely because they included false news that are harder to fact-check (and not typically fact-checked) or because the news are less false than the typical fact-checked false news. Yet, more work is needed to investigate whether the skepticism bias documented here is due to the selection of fact-checked false news or to something else. This highlights the importance of news selection in misinformation research: Researchers need to think carefully about what population of news they sample from, and be clear about the generalizability of their findings^{42,62}. Overall, our results are informative about people’s ability to spot fact-checked false news, and about their doubts towards mainstream true news. However, our results also suggest that people discern worse for more representative samples of misinformation news. More research designed to overcome news selection bias is needed to provide a solid account of how much worse.”

Reviewer 3: - *The ability to spot false news surely depends on the specific news. It would be easy to devise an experiment with a selection of true and false headlines on esoteric subjects where participants will certainly lack the required knowledge to spot the difference. It would be similarly easy to devise an experiment that is all but certain to find a large difference. I’m sure most of this work is done in good faith, but don’t we know enough about (unconscious) researcher bias to consider this aspect of the experiments? I’m reassured to see that some studies used a random selection of true headlines (Appendix G), but the selection of false headlines remains a bit of black box. The implication seems to be: “given the selection of headlines that were chosen by researchers in this field, most people do not lack the skills to spot false news”*

Authors: This is a central point and the reviewer is right that our results only hold for the given set of headlines selected by researchers. Yet, we believe that it is unlikely that discernment is entirely attributable to researchers’ news selection. First, our meta-analysis includes 2167 unique headlines, and out of the 303 effect sizes, 298 are positive (of the positive estimates, 3 have a confidence interval that includes 0, as does 1 of the negative estimates). Second, selecting easy to discern true and false news often goes against the researchers’ incentives: Many studies tested interventions against misinformation, such as media literacy tips, which are generally hypothesized to increase discernment. Therefore, researchers have an interest to avoid floor effects for false news accuracy ratings, and ceiling effects for true news accuracy ratings, otherwise there is no room for improvement. In other words, if anything, this incentive should have led to the selection of hard to discern headlines.

We lay out this “disincentive” of artificially selecting for discernment when discussing the possibility of publication bias in our manuscript:

“However, a priori we did not expect publication bias to be present because our variables of interest were not those of interest to the researchers of the original studies: Researchers generally set out to test factors that alter discernment, and not the state of discernment in the control group. No study measured skepticism bias in the way we define it here.” (p.20)

When starting the project, we shared the concern of the reviewer and decided to code whether a study selected news items based on a pre-test that included accuracy ratings. Note that this is not a variable capturing whether *any* pre-test has been run: e.g., many papers on political concordance ran pre-tests to identify the political slant of news headlines, but did not ask about accuracy, which is not relevant for the question at hand.

We identified 13 papers in which researchers pre-tested their news items and measured accuracy, or explicitly stated that they selected their news taking accuracy into account. In the table below, we provide an overview of these studies and the direction of their accuracy selection bias. Note that the details of these pre-tests are often not very clear (e.g., Peren et al. 2023 write simply writes that “pretested the headlines among Ph.D. students to ensure sufficient variation in response.”) and their scope is sometimes very limited (e.g. Modirrousta-Galian et al. 2023 excluded one news item out of 50 for eliciting floor effects).

In total, among papers who explicitly stated selection with regard to accuracy 7 papers can be argued to have tried maximizing discernment, while 4 papers can be argued to have done the opposite. For the bulk of studies who didn’t do (or report) a pre-test, we believe incentives should have favored minimizing discernment in control conditions.

Informally, and anecdotally, we talked to a few researchers in the field of misinformation about their practices of headline selection and most of them told us not to pre-test them, and those who pre-test them reported doing so based on partisanship (e.g., to make sure that the pro-democrats headlines were indeed perceived as favorable to democrats by participants).

In conclusion, while we advocate for automated news selection based on objective factors (e.g., popularity), we do not believe that we observe discernment that was artificially created by researchers—that is, beyond the fact that researchers relied on fact-checked false news.

ref	reference	selection
Altay_2022_b	Altay, S., Nielsen, R. K., & Fletcher, R. (2022). The impact of news media and digital platform use on awareness of and belief in COVID-19 misinformation [Preprint]. PsyArXiv. https://doi.org/10.31234/osf.io/7tm3s	reduce discernment, i.e. avoid floor and ceiling effects (however, no pre-test)

Roozenbeek_2022	Roozenbeek, J., Maertens, R., Herzog, S. M., Geers, M., Kurvers, R., & Sultan, M. (2022). Susceptibility to misinformation is consistent across question framings and response modes and better explained by myside bias and partisanship than analytical thinking. Judgment and Decision Making , 17(3), 27.	Used items from Maertens et al. 2021
Maertens_2021	Maertens, R., Götz, F. M., Schneider, C. R., Roozenbeek, J., Kerr, J. R., Stieger, S., McClanahan, W. P., Drabot, K., & Linden, S. van der. (2021). The Misinformation Susceptibility Test (MIST): A psychometrically validated measure of news veracity discernment [Preprint]. PsyArXiv. https://doi.org/10.31234/osf.io/gk68h	maximize discernment (however, the authors claim that at the same time they selected for preserving high variability)
Sultan_2022	Sultan, M., Tump, A. N., Geers, M., Lorenz-Spreen, P., Herzog, S., & Kurvers, R. (2022). Time Pressure Reduces Misinformation Discrimination Ability But Not Response Bias. PsyArXiv. https://doi.org/10.31234/osf.io/brn5s	reduce discernment, i.e. avoid floor and ceiling effects
Rathje_2023	Rathje, S., Roozenbeek, J., Van Bavel, J.J. et al. Accuracy and social motivations shape judgements of (mis)information. Nat Hum Behav (2023). https://doi.org/10.1038/s41562-023-01540-w	defined political slant based on accuracy ratings (e.g. high accuracy ratings from Republicans, but low accuracy ratings from Democrats = pro-republican news); since it evens out for both groups, not relevant here
Arechar_2022	Arechar, A. A., Allen, J., Berinsky, A. J., Cole, R., Epstein, Z., Garimella, K., Gully, A., Lu, J. G., Ross, R. M., Stagnaro, M. N., Zhang, Y., Pennycook, G., & Rand, D. G. (2023). Understanding and combatting misinformation across 16 countries on six continents. Nature Human Behaviour , 7(9), 1502–1513. https://doi.org/10.1038/s41562-023-01641-6	maximize discernment
Altay_2023	Altay, S., Lyons, B., & Modirrousta-Galian, A. (2023). Exposure to Higher Rates of False News Erodes Media Trust and Fuels Skepticism in News Judgment. https://doi.org/10.31234/osf.io/t9r43	maximize discernment (however, comparing pre-test and main study results, they find very similar discernment)

Ross_2018	Ross, B., Heisel, J., Jung, A.-K., & Stieglitz, S. (2018). Fake News on Social Media: The (In)Effectiveness of Warning Messages.	reduce discernment, i.e. avoid floor and ceiling effects (however, while they state that intention, it seems no study has been excluded for that reason)
Luhring_2023	Luhring et al. (2023) Emotions in misinformation studies: Distinguishing affective state from emotional response and misinformation recognition from acceptance	maximize discernment
Modirrousta-Galian_2023	Modirrousta-Galian, A., Higham, P. A., & Seabrooke, T. (2023, May 9). Wordless Wisdom: The Dominant Role of Tacit Knowledge in True and Fake News Discrimination. Retrieved from psyarxiv.com/2gubk	reduce discernment (but note that only one item of 50 items was excluded for that reason, a fake news item that showed floor effects)
Peren_2023	Peren Arin, K., Mazrekaj, D., & Thum, M. (2023). Ability of detecting and willingness to share fake news. Scientific Reports, 13(1), 7298.	maximize discernment (however, no pre-test)
Winter_2024	Winter, S., Valenzuela, S., Santos, M., Schreyer, T., Iwertowski, L., & Rothmund, T. (2024). (Don't) Stop Believing: A Signal Detection Approach to Risk and Protective Factors for Engagement with Politicized (Mis)Information in Social Media.	unclear; they measure discernment in a pre-test don't say if it mattered for selection
Guess_2024	Guess, A., McGregor, S., Pennycook, G., & Rand, D. (2024). Unbundling Digital Media Literacy Tips: Results from Two Experiments. OSF. https://doi.org/10.31234/osf.io/u34fp	unclear; a pre-test included an accuracy measure, but might not have played a role for selection

Reviewer 3:

Skepticism bias

- The same issues of ecological validity also apply to skepticism. Prompting participants to rate the veracity of headlines alerts them to the possibility of fake headlines and may make them more skeptical than they would otherwise be. The problem here is greater, because here the authors do interpret the result as a reflection of what news consumers would do in real life. I think the paper would benefit from an acknowledgment of the fact that the skepticism bias may have been inflated by the experimental context. I do appreciate the discussion of the danger posed by excessive skepticism (p12) and by interventions that reduce trust in false and true news alike (p13). But I also feel that the skepticism bias effect found here should not be overstated.

Authors: Of course, we agree with the reviewer and we extended the section in which we discuss potential experimenter effects:

- “First, participants evaluated the news stories in artificial settings that do not mimic the real-world. For instance, the mere fact of asking participants to rate the accuracy of the news stories may have increased discernment by increasing attention to accuracy³³. When browsing on social media, people may be less discerning (and perhaps less skeptical) than in experimental settings because they would pay less attention to accuracy³⁷.” (p.14)

We are not convinced that prompting participants to evaluate the accuracy of news should induce a skepticism bias rather than a truth bias: in most scales, participants were asked how ‘accurate’ or ‘true’ the headlines were which may induce a truth bias as the wording focuses on positive characteristics, whereas one might expect a skepticism bias if the wording was more negative, e.g., ‘inaccurate’, ‘false’, ‘fabricated’, etc. However, this is ultimately an empirical question, and we would bet that the most likely effect to expect is a null. (We are aware that the accuracy prompts mostly work by reducing the sharing of false news and not by increasing the sharing of true news, even though the wordings of the prompts are mostly positive. However, it is not clear to what extent it would translate into accuracy ratings given that the effect of the prompt cannot be measured on accuracy.)

Throughout the manuscript, we have toned down the effect of the skepticism bias, and made clear that the effect is small and likely a result of our focus on fact-checked false news:

“In conclusion, we found that in experimental settings, people are able to discern mainstream true news from fact-checked false news, but when they err, they tend to do so on the side of skepticism more than on the side of gullibility (although the effect is small and likely contingent on false news selection).” (p.15)

Reviewer 3: - *Calculating the differences between the mean scores and the end-point of the scales seems like an acceptable way of assessing skepticism bias, but it does assume that only veracity judgments are assessed, and not, for instance, certainty. But perhaps participants held two thoughts in their head: how accurate they thought the headline was and how certain they were about this. Even for headlines that they accept as true, participants may have been reluctant to indicate the highest score on the veracity scale. After all, can you ever be 100% sure that something is true? Choosing the most extreme score (the lowest score) is easier in the case of headlines that participants think are obvious nonsense. I’m not sure that this indicates a worrying rejection of true headlines. Rather, it may be a simple acknowledgment that some things are definitively untrue, whereas there are not a lot of things that you can be certain are definitively and irrevocably true. Could this also be the reason why studies using 7-point scales have such lower skepticism bias? Because participants had more opportunity there to indicate the veracity of true headlines while shunning the most extreme answer? I realize this is not the only possible explanation of this moderator effect, but I do feel the moderation by the type of accuracy scale (p9) raises questions about the employed methodology. I wonder what would happen if the analysis would be run on the average percentage of true headlines that participants rate as true (higher than the midpoint of the scale) minus the average percentage of false headlines that participants rate as false. Perhaps I misunderstand, but I think this would be similar to the studies using binary scales, and for those studies the effect is also much smaller (p9). Is skepticism bias a direct result of the 4-point scale?*

Authors: We thank the reviewer for raising several interesting questions regarding response scales. First, we now mention the point that true news might simply be less obviously true in the discussion:

“The skepticism bias documented here may stem from the fact that it’s easier for something to be false than for something to be true. Falsifying a statement is easier than confirming it: there only needs to be one black swan to falsify the statement that all swans are white whereas confirming this statement requires much more effort. This may explain why participants were more eager to classify news stories as false rather than true. Yet, it does not explain why the literature on interpersonal communication typically finds a truth bias and why people tend to show an acquiescence bias rather than a rejection bias.”

Note also that we are now clearer about the fact that the skepticism bias is very likely due to our focus on fact-checked false news (see Appendix G), although more work is needed to confirm this.

Second, we have conducted new analyses on the effect of the scales, as suggested by the reviewer. We show that the skepticism bias is not a direct result of the 4-point scale, because even on binary scales, we do find a positive (but smaller) skepticism bias (see moderator analyses and Appendix D). And yet, as we discuss in Appendix D, it seems likely that skepticism bias stems partly from mis-classifications, partly from degrees of confidence. We establish this conclusion based on analyses of a subsample of studies that we could extract individual-level data on and that used Likert scales. For this sub-sample, we compared the original likert scale version with a version where we dichotomized answers:

- “In a first step, to provide a test, we use a subset of studies we have individual-level data on, and collapse Likert scale response into dichotomous answers. For example, on a 4-point scale, we coded responses of 1 and 2 as not accurate (0) and 3 and 4 as accurate(1). For scales, with a mid-point (example 3 on a 5-point scale), we coded midpoint answers as 'NA'. We find a skepticism bias with the original Likert scale version (see also Appendix C), but not with the dichotomous version. In a second step, we look at studies we have individual-level data on and which use binary answer scales. For these studies, we do find both positive discernment and positive skepticism bias, although smaller estimates than our overall meta-analytic averages. We replicate this finding when adding the dichotomized version of the likert scale studies from the first test. We further show that these results hold when using more appropriate summary statistics for binary outcomes, namely (log) odds ratios. Taken together, this suggests that skepticism bias stems partly from mis-classifications (the observed skepticism bias in binary response studies), but partly from degrees of confidence (the difference between likert-version and collapsed binary version). On average, people tend to (i) classify true news as false more often than false news as true and (ii) even when classifying equally well for both and true news, they rate true news as less extremely accurate than false news as inaccurate, suggesting lower confidence in their accuracy answers for true news.” (Appendix D)

Note that in the cited section, we conclude that skepticism bias partly stems from “degrees of confidence”, not degrees of (objective) news accuracy. The reviewer raises the question of whether news stories are simply more false than true. If news are objectively either true or false, then Likert scale accuracy ratings reflect nuances in participants' confidence in their judgment.

But if news veracity lies on a continuum between true and false, then Likert scale ratings could also reflect actual differences in degrees of accuracy of news. We adopt the paradigm of all here included studies, which is to assume that news veracity can be established in the binary, true vs. false classification. We believe that this is reasonable, considering that the studies used only headlines or at most a short lede, and not longer news stories with more room for ambiguity.

Even assuming objective ambiguity of news' veracity, our principle conclusions remain unchanged: Binary response scales should be robust to modest degrees of ambiguity. Our moderator analyses show that our findings are true also for studies using binary response scales, despite smaller effect sizes.

Reviewer 3: Minor issues

- Crowdsourced fact-checking initiatives are mentioned rather prominently in the Discussion as well as the Abstract, but to me that feels like a rather big leap. I would personally prefer a more in-depth discussion of what these results mean (see above).

Authors: We have greatly extended the discussion, and engage in more in-depth discussion of what the results mean. However, we do not see crowdsourced fact-checking as a particularly big leap: for crowdsourced fact-checking to work people need to be able to distinguish true from false news, and many commentators have expressed skepticism about crowdsourced fact-checking because they did not believe that people were indeed able to do so. We also believe that it is an interesting point of comparison given that participants were mostly asked to evaluate fact-checked false news. We have made the relevance of this comparison clearer (in light of the limitations that comes with focusing on fact-checked false news):

- “Second, the fact that people can, on average, discern true from false news lends support to crowdsourced fact-checking initiatives. While fact-checkers cannot keep up with the pace of false news production, the crowd can, and it has been shown that even small groups of participants perform as well as professional fact-checkers^{39,40}. The cross-cultural scope of our findings suggests that these initiatives may be fruitful in many countries across the world. In every country included in the meta-analysis, participants on average rated true news as more accurate than false news (see Appendix ??). Our results are also informative for the work of fact-checkers. In recent years, fact-checking organizations such as PolitiFact have mostly focused on debunking false news at the expense of confirming true news⁴¹. Yet, we show that people also need help to identify true news as true. Moreover, since people are quite good at discerning true from false news, fact-checkers may want to focus on headlines that are less clearly false or true. However, we cannot rule out that people’s current discerning skills stem in part from the work of fact-checkers.” (p.12)

Reviewer 3: *- On page 4, the authors say that they only included studies with neutral control conditions. Does this mean that only the data from the control conditions was used? I think examples would be nice of the kinds of studies that were included, so that readers who are not familiar with this literature get a sense of the experiments.*

Authors: In almost all cases, a control condition corresponds to what we call a “neutral” control condition, i.e. one that is comparable across papers and free from any design feature that could have additionally affected ratings of our outcomes in comparison to other papers. To provide an example, we added a footnote on p. 4, where we write:

- “For example, (30), among other things, test the effect of an interest prime vs. an accuracy prime. A neutral control condition - one that is comparable to those of other studies - would have been no prime at all. We therefore excluded the paper.”

Reviewer 3:- *On a similar note, when did the same sample contribute several discernment effect sizes (p7)? Is this for instance when politically discordant and concordant groups were used and different effect sizes were included? Or is the explanation the same as for the skepticism effect size (p8).*

Authors: We extended the footnote to make this clearer:

- “Sometimes, a sample provided several effect sizes, for example when separate accuracy ratings are available by news topic, or when follow-up studies were conducted on the same participants. A common case where a sample provides several effect sizes is when participants rated both politically concordant and discordant news. In this case, if possible, we entered summary statistics separately for the concordant and discordant items, yielding two effect sizes (i.e. two different rows in our data frame). We account for the resulting hierarchical structure of the data in our statistical models.” (p.4)

Reviewer 3:- *For the studies using a binary scale, am I correct in inferring that this basically comes down to calculating the average proportion of times participants correctly identified a false headline minus the average proportion of times participants correctly identified a true headline?*

Authors: Not exactly (but we have a comparison using “correct identification as a measure in Appendix C). On a binary scale, we can break responses down into four categories:

- True Positives (TP): Instances where true news is correctly identified as true.
- False Positives (FP): Instances where false news is incorrectly identified as true.
- True Negatives (TN): Instances where false news is correctly identified as false.
- False Negatives (FN): Instances where true news is incorrectly identified as false.

Discernment is average true news as accurate minus average false news as accurate ratings. This translates to:

$$\text{Discernment} = (TP / (TP + FN)) - (FP / (FP + TN))$$

(In other words, the true positive rate among true news minus the false positive rate among false news)

Reviewer 3:- *Are the difference scores reported in the Moderators section meta-regression weights? Why not (also) report the composite effect sizes for the subgroups?*

Authors: We’re sorry for the ambiguity. The “Delta” values are really just the coefficients of the meta-regressions. Since these coefficients (or beta’s) designate differences between categories, we thought “Delta” would make the interpretation more intuitive. We have clarified this in the introductory paragraph of the moderator section:

“Following the pre-registered analysis plan, we ran a separate meta regression for each moderator by adding the respective moderator variable as a fixed effect to the multilevel

meta models. We report regression tables and visualizations in Appendix B. Here, we report the regression coefficients as "Delta"s, since they designate differences between categories. For example, in the moderator analysis of political concordance on skepticism bias, "concordant" marks the baseline category. The predicted value for this category can be read from the intercept (-.2). The "Delta" is the predicted difference between concordant and discordant (.78). To obtain the predicted value for discordant news, one needs to add the "Delta" to the intercept ($-.2 + .78 = .58$).” (p.9)

Reviewer 3:- *The interpretation of the effect of political concordance seems off to me (p13). The authors state that participants became more skeptical when the headlines were discordant, but the opposite is also true: they became more gullible when the headlines were concordant. Suggesting that participants became “excessively skeptical” for discordant but not “excessively gullible” for concordant headlines seems slightly misleading to me. That’s simply a result of the starting position being a slight bias toward skepticism. A bias for which the interpretation is not entirely unproblematic, I might add (see above).*

Authors:

Intellectually this was a very challenging point. While at first we agreed with the reviewer and decided to revise our interpretation of the finding, after long reflections we came to the conclusion that this interpretation is not correct.

The ‘starting’ position mentioned by the reviewer is not a neutral position and is actually an ‘ending’ position: it is the average of concordant and discordant headlines, as well as headlines not divided by concordance. Taking headlines not divided by congruence as a starting point is problematic: the average of such categories may simply be the average of concordant vs discordant headlines. For example, in a study on Covid-19 news, an anti-vaccine headline would be concordant with a vaccine skeptic’s world views. Political news is the only news category divided by concordance and if we were to divide the other news topics by concordance, such as ‘vaccine-hesitant’ vs. ‘vaccine-supporters’, we may observe the exact same effects as for politics.

On a side note, we believe that the average skepticism documented in the meta-analysis might be, to a large extent, a result of the fact that people are very skeptical of incongruent news. Unfortunately, as we mention above, we cannot divide most news items into categories of congruence and thus cannot test this hypothesis. In the discussion, we now mention that future studies should investigate this issue further.

“Future studies should investigate whether the effect of congruence is specific to politics or if it holds across topics, and compare it to a baseline by including neutral items.” (p.14)

We attempted to mention the reviewer’s interpretation in the discussion but because we also needed to highlight the limits of this interpretation it took a lot of space in the discussion. However, if the reviewer or the editor deems this interpretation particularly important to mention, and/or is not convinced by our argument, we can integrate it back.